# Cholinergic basal forebrain degeneration due to sleep-disordered breathing exacerbates pathology in a mouse model of Alzheimer's disease

Lei Qian [1,2,3], Oliver Rawashdeh [3], Leda Kasas[3], Michael R. Milne[1,2,3], Nicholas Garner[3], Kornraviya Sankorrakul[3,4], Nicola Marks[1], Matthew W. Dean[3], Pu Reum Kim[3], Aanchal Sharma[1], Mark C. Bellingham[3] & Elizabeth J. Coulson [1,2,3]✉

Although epidemiological studies indicate that sleep-disordered breathing (SDB) such as obstructive sleep apnea is a strong risk factor for the development of Alzheimer's disease (AD), the mechanisms of the risk remain unclear. Here we developed a method of modeling SDB in mice that replicates key features of the human condition: altered breathing during sleep, sleep disruption, moderate hypoxemia, and cognitive impairment. When we induced SDB in a familial AD model, the mice displayed exacerbation of cognitive impairment and the pathological features of AD, including increased levels of amyloid-beta and inflammatory markers, as well as selective degeneration of cholinergic basal forebrain neurons. These pathological features were not induced by chronic hypoxia or sleep disruption alone. Our results also revealed that the cholinergic neurodegeneration was mediated by the accumulation of nuclear hypoxia inducible factor 1 alpha. Furthermore, restoring blood oxygen levels during sleep to prevent hypoxia prevented the pathological changes induced by the SDB. These findings suggest a signaling mechanism whereby SDB induces cholinergic basal forebrain degeneration.

The causes of idiopathic dementia, of which Alzheimer's disease (AD) is the largest subgroup, are not clear. The leading theory of AD is based on familial forms of the condition in which the cleavage of amyloid precursor protein (APP) is altered, favoring production of the neurotoxic peptide amyloid-beta (Aβ)[1,2]. However, the physiological triggers that lead to change in APP metabolism, or that result in Aβ accumulation, are poorly defined[2]. Furthermore, Aβ accumulation alone is not sufficient to induce cognitive decline in the elderly. Elucidating the etiology of idiopathic AD is therefore crucial to providing efficacious

early intervention and/or treatment for the majority of people who develop AD.

Obstructive sleep apnea (OSA), the most prevalent form of sleep disordered breathing (SDB), is a strong epidemiological risk factor for the development of dementia[3]. It affects more than 50% of the elderly adult population[4] and occurs due to the collapse of the upper airways, particularly during rapid eye movement (REM) sleep; this impedes airflow and requires a brief arousal in order to re-activate airway tone[5,6]. OSA has been associated with both an earlier age of AD onset,

[1]Queensland Brain Institute, The University of Queensland, Brisbane, QLD 4072, Australia. [2]Clem Jones Centre for Ageing Dementia Research, Queensland Brain Institute, The University of Queensland, Brisbane, QLD 4072, Australia. [3]School of Biomedical Sciences, Faculty of Medicine, The University of Queensland, Brisbane, QLD 4072, Australia. [4]Research Center for Neuroscience, Institute of Molecular Biosciences, Mahidol University, Salaya, Thailand. ✉e-mail: e.coulson@uq.edu.au

more rapid cognitive decline[5,7,8] and an increased Aβ burden[9–11]. However, SDB is not widely recognized as one of the factors that increases the risk of developing AD[12].

Although the reasons for the epidemiological association between SDB and AD are unclear, there are a number of non-mutually exclusive possibilities. For example, longitudinal studies suggest that the hypoxia which results from SDB causes neurodegeneration, thereby leading to cognitive decline[6,13]. Similarly, exposure of mice to intermittent hypoxia can exacerbate the accumulation of Aβ[14] and cause neurodegeneration[15]. However, a more recent hypothesis is that disrupted sleep slows the clearance of Aβ from the brain interstitial space through the glymphatic system[16]. Sleep is also considered fundamental for learning and memory consolidation, and poor sleep quality leads to failed memory consolidation[17]. Therefore SDB-induced sleep disruption could contribute to cognitive impairment and eventually dementia.

The reasons for the increased risk that SDB creates are difficult to determine in humans, even in longitudinal studies, due to the high comorbidity rates of other risk factors such as diabetes and cardiovascular disease in people suffering from the condition[12]. Current animal models of SDB are also limited to a single feature such as chronic hypoxia, intermittent hypoxia induced artificially by rapid cycling of air in a small chamber with varying oxygen concentrations, or sleep disruption. In order to understand the mechanisms linking AD with OSA we developed a method to model SDB in mice in which breathing and sleep were disrupted coincidently but in the absence of comorbidities. We then used this model to investigate mechanisms by which SDB affects the etiology of key hallmarks of AD.

## Results

### Urotensin 2-saporin toxin specifically lesions cholinergic mesopontine neurons

Mesopontine tegmentum (MPT) neurons innervate brain regions that are important in REM sleep, such as thalamocortical regions and the basal forebrain[18], and are strongly implicated in initiating and maintaining REM sleep[19–21]. Cholinergic MPT (cMPT) neurons also project to the hypoglossal motor nucleus, which controls the tongue muscles, as well as to other brainstem areas[18]. Respiratory activity in upper airway muscle motor neurons is driven by the brainstem respiratory central pattern generator[22], and the tongue muscles are important upper airway dilator muscles during inspiration[23]. We, therefore, reasoned that cMPT lesioning would result in upper airway collapse, particularly during REM sleep when acetylcholine activity is highest[24], providing a mouse model of SDB/OSA.

cMPT neurons selectively express the urotensin 2 receptor, and respond to its ligand, urotensin-2 peptide[25–27]. To selectively ablate cMPT neurons we used a ribosomal inactivating saporin-conjugated urotensin-2 peptide (UII-SAP) and targeted the lateral dorsal tegmentum (LDT) region rather than the pedunculopontine nucleus (PPN) of the MPT, which is known to mediate locomotion or reward[28]. Direct injection of the UII-SAP toxin into the brainstem induced a unilateral specific loss of cholinergic neurons within the MPT after 2 weeks (Fig. 1; the dose response of UII-SAP is shown in Supplementary Fig. 1A). Compared to a control injection of rabbit-IgG-saporin (IgG-SAP) or blank-saporin (Blank-SAP) which is conjugated with a non-targeted peptide consisting of similar, comparable materials, the number of choline acetyltransferase (ChAT)-positive neurons in the LDT was significantly reduced in the UII-SAP-injected mice (Fig. 1A, B), with a smaller but significant loss of ChAT-positive neurons also being detected in the PPN. A similar loss of cLDT neurons was observed when UII-SAP was injected via the intracerebral ventricle (Fig. 1C), indicating that the reduced number of ChAT-positive neurons was not due to damage at the injection site. However, there was no loss of calbindin-positive GABAergic neurons in the LDT (Fig. 1D) or of cholinergic hypoglossal motor neurons (Fig. 1E). cMPT-lesioned mice displayed no

significant change in body weight (see Source Data), distance traveled (Fig. 1F) or anxiety/exploration (Supplementary Fig. 1B) in an open field test, or rotarod performance (Fig. 1G), relative to control-lesioned mice.

### cMPT lesions result in sleep-disrupted breathing

Next, we used unrestrained whole-body plethysmography during the lights on period to determine the effect of cMPT lesion on the breathing pattern of lesioned and non-lesioned mice of each sex during their sleep and wake periods as measured by electro-encephalography (EEG). We hypothesized that if there was reduced air flow due to resistance of the upper airways of lesioned mice during sleep, peak inspiration volume (PIF) would be reduced and time for inspiration (EIP) increased[29], and that the time between breaths would be increased, as occurs in humans with OSA. Due to the possible influences of sex, sleep-wake, and lesion, and the number of plethysmography variables measured that make up a breathing cycle, we have summarized the combined data (Fig. 2A–D) in which the average measures for male (Fig. 2E) and female (Sup Fig. 2B) mice were used to draw the stylized breaths. Importantly, during wakefulness, we found no significant differences in the average value of any of the breathing parameters between control and cMPT-lesioned mice of either sex (Fig. 2A, C, E, and Source Data). In contrast, a significant effect of the lesion was found on the above breathing measures during sleep ($p < 0.001$, sleep-wake-paired, two-way ANOVA).

As expected, the average frequency of breaths (f) decreased during sleep compared to wake periods in control animals (Fig. 2C–E). However, breath frequency was also significantly different between males and females ($p = 0.0109$, sleep-wake paired, two way ANOVA). Surprisingly, the breath frequency during sleep of lesioned mice was on average lower than that of controls and not significantly different from the breath frequency during wakefulness (Fig. 2A–E). However, in lesioned mice of both sexes, the time between breaths during sleep, irrespective of mean frequency, was significantly more variable than that of sleeping control mice ($P < 0.0001$; Levene test of variance; Fig. 2C). In humans, an apnea is the cessation of airflow for 10 s, which is the time in which a person would typically take three breaths (with a frequency of 12–20 breaths/min while asleep). During sleep, control mice average 5 breaths every 2 s (a rate of 150 breaths/min). However, cMPT-lesioned mice were significantly more likely to have adjoining epochs of reduced breath frequency (<100 breaths/min; $p = 0.020$, one-tailed unpaired t-test with Welch correction), with 54 and 17% of low frequency breathing epochs lasting longer than 4 s in cMPT-lesioned and control mice, respectively. Interestingly, there was a trend for the long low-frequency breathing periods to be followed by an awake epoch in the lesioned mice, whereas there was no obvious association between low frequency breathing epochs and sleep/wake transitions in the control mice.

In both control and cMPT-lesioned mice, the average PIF was reduced during sleep (Fig. 2D). However, in the lesioned mice the average EIP during sleep was shorter than that of control mice, being equivalent to their average EIP during the wake phase. Nonetheless, during sleep, PIF and EIP were more highly variable in the cMPT-lesioned mice than control mice. This indicates that, while on average inspiration flow was slower and took longer, there were both very large and very small peak flow measures and inspiration times in cMPT lesioned mice during sleep.

As for control mice during sleep, PEF was reduced in lesioned mice of both sexes during sleep—indicative of a slower exhalation period. However, the breathing patterns during exhalation of male (Fig. 2E) and female (Supplementary Fig. 2) mice differed independently of sleep, particularly in average measures of total time to exhale (EEP; $p = 0.0212$ sleep-wake-paired, two way ANOVA) and for the point in expiration where the peak flow occurs as a fraction of total time to expire (Rpef, $p = 0.051$); the control male mice had significantly larger

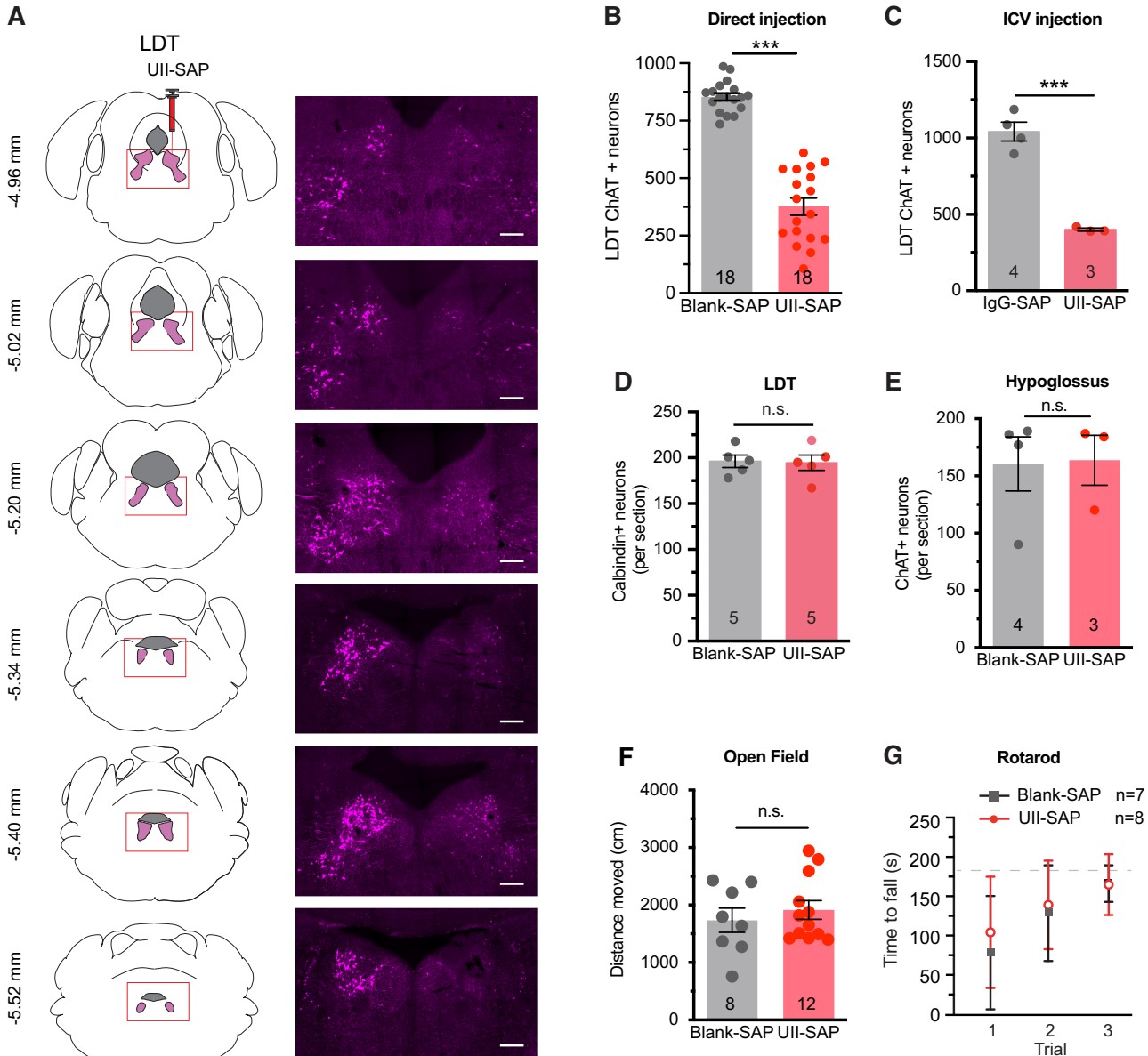

**Fig. 1 | Urotensin II-saporin induces specific lesions of cholinergic neurons at mesopontine tegmentum. A** Diagrams and photomicrographs of coronal sections of the brainstem, the right column being immunostained for ChAT-positive neurons within the laterodorsal tegmental nucleus (LDT) following unilateral direct injection of UII-saporin (UII-SAP) into the right mesopontine tegmentum (MPT). Scale bar = 200 μm. Images are representative of *N* = 3 animals. **B** Direct injection bilateral of UII-SAP into the MPT reduces the number of ChAT-positive neurons within the LDT compared with Blank-SAP injections (*P* < 0.0001). **C** Intraventricular injection of UII-SAP reduced the number of ChAT-positive neurons within the LDT compared to control unconjugated saporin injections (Blank-SAP) (*P* = 0.0003). **D** Direct injection of UII-SAP into the LDT does not affect the number of calbindin-

positive GABAergic neurons in the LDT compared to Blank-SAP injection (*P* = 0.8857). **E** The number of ChAT-positive hypoglossal motor neurons per section following injection of UII- SAP or Blank-SAP is not different (*P* = 0.9470). **F** The distance traveled by UII-SAP- and Blank-SAP-injected mice in the center area of the open field test, which was not different between conditions (*P* = 0.8313). **G** Time spent on the Rotarod in three successive trials lasting up to 3 min each is not different between UII-SAP- and Blank-SAP-injected mice. Comparisons by Students' unpaired two-tailed t-test; ****P* < 0.0001, n.s., non-significant. Results are presented as mean ± s.e.m. Each data point represents an individual animal. Source data are provided in the Source Data file.

averages of these measures during sleep than females, and as the values for males but not females were significantly reduced in the lesioned groups, a significant interaction with sex and sleep state in EEP was observed (*p* = 0.0157 sleep-wake-paired, two way ANOVA). Nonetheless, the average EEP for sleeping lesioned mice was significantly less than sleeping control mice (*p* < 0.0491, paired two-way ANOVA), and was equivalent to that of awake mice (*p* > 0.999). However, in the male lesioned mice, EEP during sleep was significantly more variable than that of control sleeping animals (*P* < 0.0001; Levene test of variance; Fig. 2C). Despite this, the time for 65% of

exhalation to be reached was less variable in both male and female lesioned mice during sleep than for the control mice during sleep (*p* < 0.0001; Levene test of variance; Fig. 2C). As the EEP includes any pause between breaths, these results account for the lesioned mice taking more frequent breaths on average, and also indicates that the lesioned mice, particularly males, exhibit a highly variable length of time between the point of 65% exhalation and the start of the next inhalation.

We also noted that the breathing measures during sleep for individual mice within the lesion group were significantly different for

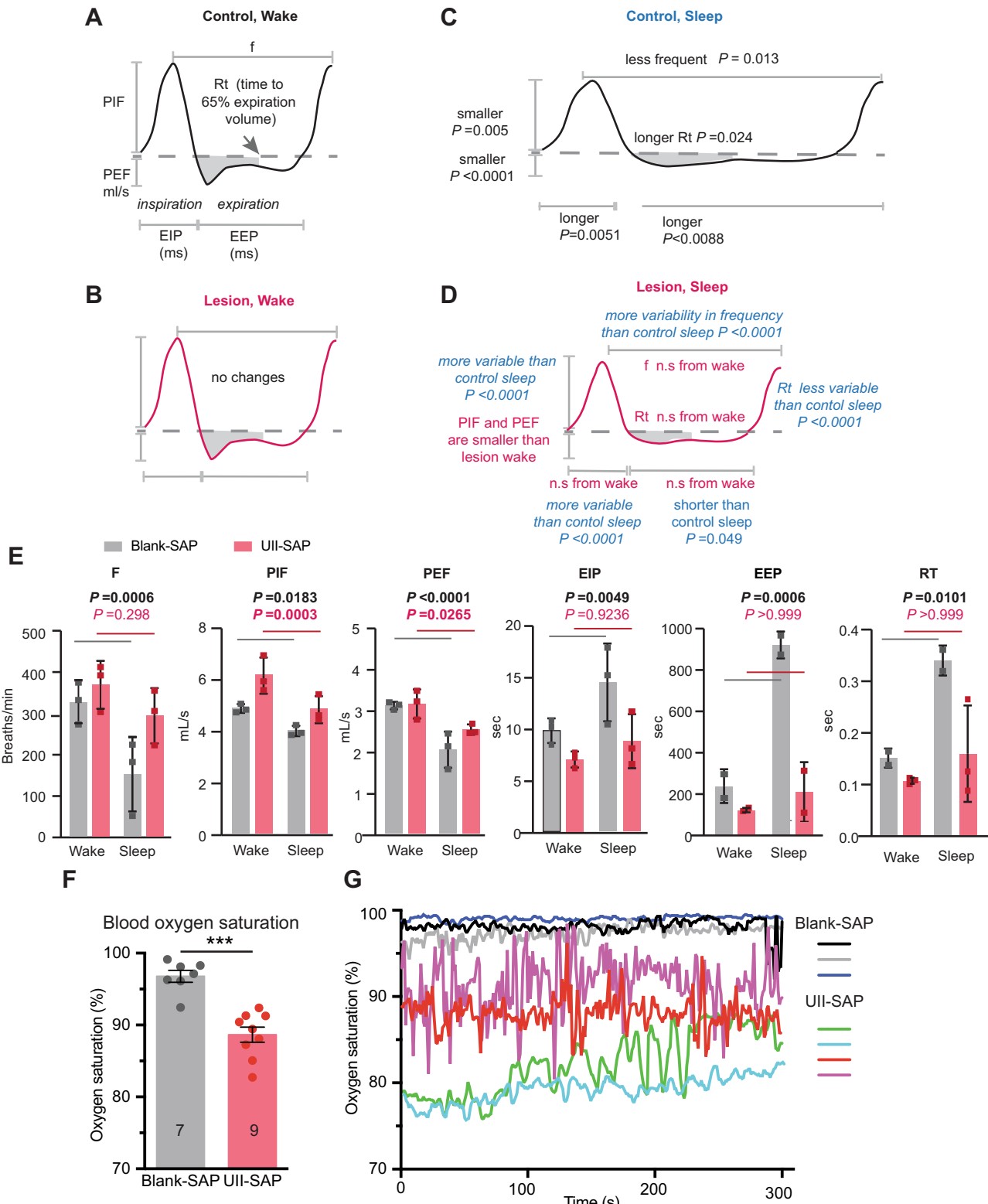

**Fig. 2 | Urotensin II-saporin treatment affects the breathing pattern and induces hypoxemia during sleep.** Stylized breathing traces based on the average measures of the polysomnography features recorded by whole body plethysmography and coincident EEG recordings in mice injected with control Blank-SAP (**A**, **C**) or UII-SAP (**B**, **D**) during wake (**A**, **B**) or sleep (**C**, **D**). Statistics refer to sleep-wake paired two-way ANOVA and Levene tests of variance of both male and female mice. **E** Average polysomnography measures for individual male mice in UII-SAP or control Blank-SAP groups (sleep-wake paired two-way ANOVA; mean ± s.d) f

frequency, PIF peak inspiration flow, PEF the peak exhalation flow, EIP time for inspiration, EEP time for exhalation, RT: **F** Average blood oxygen saturation level of unrestrained UII-SAP- and Blank-SAP-injected male mice measured during their sleep period (*P* < 0.0001, Student's unpaired two-tailed t-test; mean ± s.e.m). **G** Representative traces of blood oxygen saturation levels of male mice injected with either UII-SAP or control Blank-SAP. *P< 0.05, **P< 0.01, ***P< 0.001 ****P< 0.0001 n.s: non-significant. Each data point represents an individual animal.

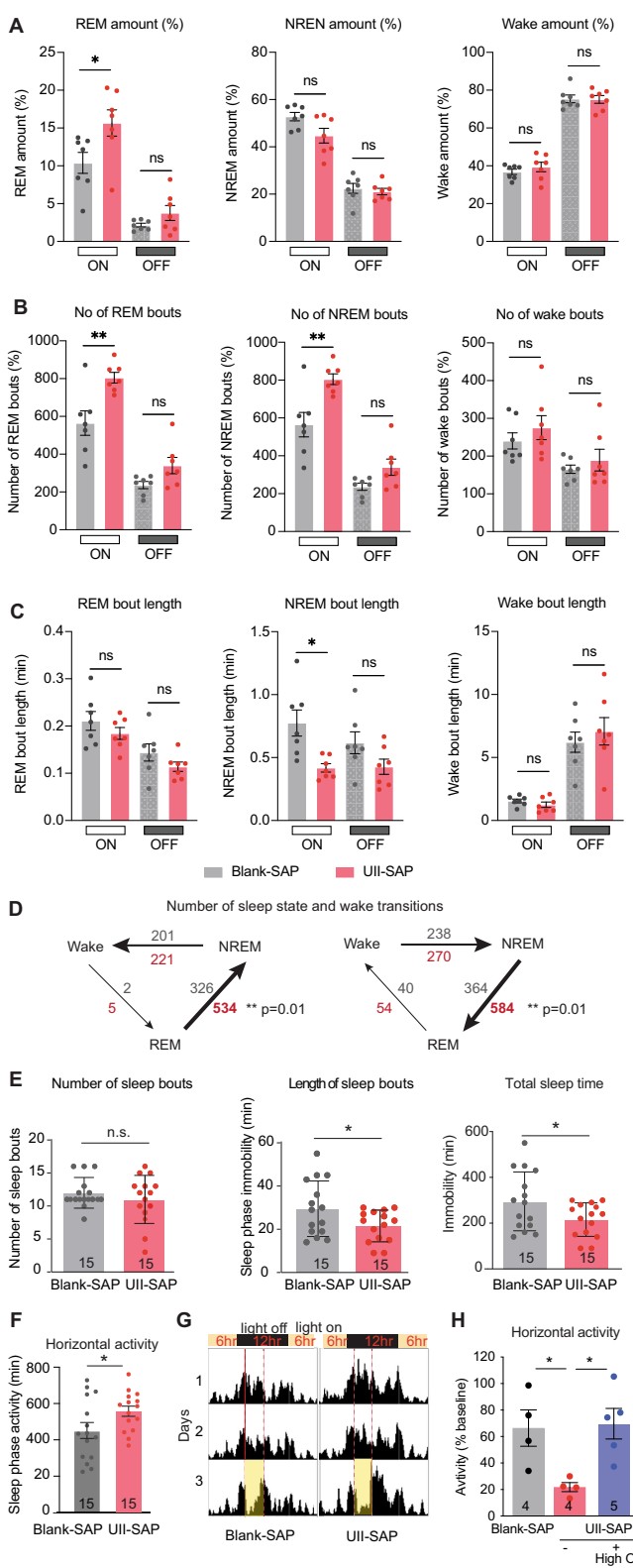

**Fig. 3 | Urotensin II-saporin treatment affects the sleeping patterns causing sleep deprivation. A** Average amount of REM, NREM, and wake sleep during lights on (sleep) and lights off (wake) periods. ($P = 0.0264$, two-way ANOVA with Tukey's multiple comparisons test). **B** Average number of REM, NREM and wake sleep bouts during lights on (sleep) and lights off (wake) periods. ($P = 0.0026$, two-way ANOVA with Tukey's multiple comparisons test). **C** Average length of REM, NREM an wake bouts during lights on (sleep) and lights off (wake) periods. ($P = 0.0145$, two way ANOVA with Tukey's multiple comparisons test). **D** The average number of transitions during the lights on period that lesioned (gray) and UTII-SAP (red) lesioned mice moved between sleep states and sleep and wake ($P = 0.010$ unpaired two-tailed t-test). **E** Average number and length of sleep bouts (periods spent inactive lasting at least 10 min, $P = 0.5023$, $P = 0.0475$, unpaired two-tailed t-test) and total time spent inactive ($P = 0.0475$, unpaired two-tailed t-test) during a 12 h light phase for the first 48 h of activity recording. **F** Total time spent inactive during the 12 h light phase in the first 48 h ($P = 0.0014$, unpaired two-tailed t-test). **G** Activity traces of a control and a lesioned mouse over 3 days of in 12:12 h light:dark cycle. On day 3 during the dark phase, mice were exposed to 3 h of 30 Lux dim light. **H** Time spent active during the 3 h low light period during the 12 h light phase on day 3 of the experiment. ($P = 0.04$). Mice placed in 40% oxygenated (high O2) for 8 h a day during the sleep period for 2 weeks starting 2 weeks after injection with UII-SAP (purple bar) did not have a sleep debt compared to UTII-SAP-injected mice subjected to normoxia. ($P = 0.0286$, two-tailed Mann–Whitney test). *$P < 0.05$, **$P < 0.01$, ***$P < 0.001$ ****$P < 0.0001$ n.s: non-significant. Results are presented as mean ± s.e.m. Each data point represents an individual animal. Source data are provided in the Source Data file.

In summary, these data demonstrate that cMPT-lesioned mice have highly variable breathing patterns during sleep that, on average, are consistent with predicted alterations in inhalation and exhalation measures that occur due to upper airway resistance[29,30], with breath to breath variability indicative of a recovery response triggered by hypoxia[31]. Most lesioned mice also displayed multi-second periods of reduced breath frequency during sleep, followed by arousal, that could be interpreted as apnea event(s). Furthermore, consistent with the most common forms of SDB, when the blood oxygen saturation of the cMPT-lesioned mice at rest was measured for 30 min during the sleep period, it was found to be between 80 and 90% SpO2, a significant reduction compared with the >95% SpO2 recorded in control mice (Fig. 2F, G). These findings indicate that the cMPT lesion affects breathing pattern, causing mild hypoxia during sleep, which is consistent with an airway obstruction.

### cMPT lesion disrupts sleep resulting in a sleep debt

We next analyzed the sleep patterns of the cMPT-lesioned mice to determine if the altered breathing pattern correlated with altered sleep states, as occurs in SDB. Male and female mice were analyzed separately. cMPT lesioning had no effect on total awake time, number of wake bouts or wake bout length during their sleep and active phases (lights on/off; Supplementary Fig. 3A). However, the number of non-REM (NREM) and REM sleep bouts were significantly increased in male mice with cMPT lesions during their sleep phase, with the duration of the NREM bouts being significantly reduced compared to control-lesioned mice (Fig. 3A–C). The number of transitions between REM and NREM sleep and vice versa were also significantly increased (Fig. 3D). Similar trends to that displayed by the male lesioned mice were observed for female lesioned mice (Supplementary Fig. 3B, C).

We next determined whether the loss of cMPT neurons resulted in a sleep debt. Male mice were individually housed on a 12:12 h light:dark cycle in a PhenoMaster cabinet and their movement was quantified using infrared motion sensors in 10 min blocks. Lesioned mice had an equivalent number of sleep bouts (inactivity-defined sleep) to control mice (Fig. 3E) but of shorter duration (≥10 min long) resulting in significantly less total sleep during the inactive phase (Fig. 3E, F), suggesting that the mice displayed disrupted sleep. To further assess the sleep quality in these mice, a 3 h dim light pulse (30 Lux) was applied during the dark (awake) phase on day 3 of the assessment to induce a

all the above measures except Rt ($P < 0.01$; sleep-wake paired, two way ANOVA). Furthermore different mice drove the variability in different breathing measures, such that on average there were no significant differences between control and lesioned animals in tidal volume or its derivatives, and not all mice displayed the extended low frequency breathing periods. This suggests that different animals may have compensated for the cMPT lesion-induced reduction in upper airway air flow through different physiological mechanisms.

sleep-conducive environment[32]. The lesioned mice displayed a sleep debt (the human equivalent of daytime sleepiness), showing a greater tendency to sleep during the light pulse (Fig. 3G, H).

To test whether the hypoxia was responsible for the disrupted sleep, we first empirically determined that a high (40%) oxygen environment restored the blood oxygen levels of cMPT-lesioned mice to >95% $SpO_2$. Starting 2 weeks after lesion surgery, male mice were treated daily with high oxygen during their 12 h sleep period for 2 weeks while in the PhenoMaster cabinet. As expected, high oxygen treatment had no effect on the extent of the cMPT lesion (Supplementary Fig. 3D). Nonetheless, the treatment restored the animals' sleep patterns, and prevented the development of a sleep debt (Fig. 3H, I), demonstrating that these phenotypes were linked to the induced hypoxia. Taken together, these findings (disrupted breathing during sleep, changed sleep patterns and induction of hypoxia) are consistent with the cMPT lesion inducing a SDB-like phenotype in mice.

## cMPT lesion exacerbates the cognitive and pathological features of AD

In order to determine whether the cMPT lesion and associated OSA phenotypes exacerbated the core features of AD (cognitive impairment, amyloid plaques, neuroinflammation, and neurodegeneration), we used the commonly studied APP/PS1 familial AD mouse model. This strain expresses mutant amyloid precursor protein (APPswe) and presenilin 1 (PS1ΔE9), leading to overproduction of Aβ, which aggregates and accumulates into amyloid plaques from around 6 months of age, resulting in cognitive impairment.

We lesioned the cMPT neurons of 8-month-old APP/PS1 mice of both sexes and tested their cognitive function two months later. No difference was observed between 10-month-old lesioned and control groups (sham-lesioned APP/PS1 mice) in terms of time spent in the center of an open field (cMPT lesioned = 46.45 ± 3.66 s, sham lesioned = 47.88 ± 5.45 s). However, cMPT-lesioned APP/PS1 mice of both sex displayed severe impairment in hippocampal-dependent learning and memory compared to age-matched unlesioned APP/PS1 mice. In the Y maze, sham-lesioned mice spent significantly more time in the novel arm of the apparatus than in the familiar arm, whereas cMPT-lesioned mice showed no preference (Fig. 4A). In the Morris water maze, APP/PS1 sham-lesioned mice learned the location of the platform during training, whereas the cMPT-lesioned mice did not (day 1 vs day = 46.45 ± 3.66 s, sham lesioned = 47.88 ± 5.45 s). However, cMPT-lesioned APP/PS1 mice displayed severe impairment in hippocampal-dependent learning and memory compared to age-matched unlesioned APP/PS1 mice. In the Y maze, sham-lesioned mice spent significantly more time in the novel arm of the apparatus than in the familiar arm, whereas cMPT-lesioned mice showed no preference (Fig. 4A). In the Morris water maze, APP/PS1 sham-lesioned mice learned the location of the platform during training, whereas the cMPT-lesioned mice did not (day 1 vs day 5: Blank-SAP, P = 0.0039; UII-SAP, P > 0.9999; Fig. 4B). In the probe trial both lesioned and sham-lesioned APP/PS1 mice had similar poor performance (Fig. 4C), likely reflecting memory impairment due to the animals' age and genotype, irrespective of lesion status. Similarly, in the active place avoidance task, neither lesioned nor control APP/PS1 mice showed significant learning during 4 days of training (Blank-SAP: day 1 vs day 4, P = 0.2286; UII-SAP: day 1 vs day 4, P = 0.9999; Fig. 4D). However, lesioned APP/PS1 mice exhibited a greater impairment in both the learning (Fig. 4D) and memory (Fig. 4E) components of this task compared to controls. Together these data indicate that cMPT neuron lesions exacerbate cognitive deficits in aged APP/PS1 mice.

To determine whether the cMPT lesions exacerbate the pathological features of AD, we assessed the number of Aβ plaques, as well as the levels of inflammation and neurodegeneration (Fig. 5A). No differences were found between male and female animals, which were

analyzed together (Sup Fig. 4). As expected, the APP/PS1 mice injected with UII- SAP at 8 months of age and sacrificed 3 months later displayed significant cLDT neuronal loss (Fig. 5B), equivalent to that of cMPT-lesioned wildtype mice (Fig. 1B). cMPT-lesioned APP/PS1 mice had significantly increased levels of soluble Aβ42 in hippocampal lysates (Fig. 5C), as well as more thioflavin S-positive and 6E10-Aβ plaques in the cortex (Fig. 5A, D, E), compared to sham-lesioned APP/PS1 mice. The levels of microglia (Fig. 5A, F, G) and activated astrocytes (Fig. 5A, H, I) in the cortex of cMPT-lesioned APP/PS1 mice were also significantly higher. We found no evidence of neurodegeneration in the hippocampus or cortex of lesioned APP/PS1 mice by measuring the width of the pyramidal neuron layers (Fig. 5J, K). However, the number of cBF neurons, a feature of AD that occurs early in the human disease but is not readily observed in the familial mouse models[33], was significantly reduced in cMPT-lesioned APP/PS1 mice compared to sham-lesioned APP/PS1 mice (Fig. 5L), as was the density of cholinergic axonal innervation to the cortical regions (Supplementary Fig. 5).

Together these results indicate that cMPT lesioning causing SDB, when coupled with a genetic drive to overproduce Aβ, can exacerbate the key pathological and cognitive features of AD in mice.

## cBF degeneration is induced by cMPT lesion

Cholinergic basal forebrain (cBF) neurons project to the entire neocortex, regulating cognitive processes including spatial navigation and associative learning[34]. These neurons are among the first to degenerate in AD, with their loss being evident in mild cognitive impairment and idiopathic AD, as well as in individuals who are subsequently diagnosed with cognitive impairment[33]. In humans, Aβ load correlates with basal forebrain atrophy and dysfunction[35–37], but it is unclear whether cBF degeneration is a cause and/or effect of Aβ accumulation. In mouse models, oligomeric Aβ can induce cBF degeneration[38,39], whereas cBF degeneration can exacerbate Aβ accumulation[40–42].

To determine the cause of cBF neuron loss observed in the APP/PS1 mice, we examined the effect of the cMPT lesioning on cBF neuron survival in male wildtype (C57Bl6J) mice over a 7-week period. Significant loss of cLDT neurons was evident by 2 weeks, consistent with the nature of the saporin toxin-induced cell death[42], with more than 50% loss of cLDT cells observed at later time points (Fig. 6A). The loss of cBF neurons occurred subsequent to the cMPT degeneration between 2 and 4 weeks after surgery (Fig. 6B), resulting in ~30% loss of cells from the medial septum, as well as the vertical and horizontal diagonal bands of Broca (Supplementary Fig. 6A, B).

Furthermore, the reduction in the number of cBF neurons significantly correlated with the size of the cMPT lesion (r = 0.5746, P = 0.0080, R2 = 0.3302). No change was observed in the number of parvalbumin-positive GABAergic neurons in the basal forebrain (Supplementary Fig. 6C). This cBF degeneration was also sufficient to cause impairment in visuo-spatial memory consolidation. No significant deficits in the Y maze (Supplementary Fig. 6D) or novel object recognition tasks (Supplementary Fig. 6E), were displayed by cMPT-lesioned wildtype mice or unlesioned mice. However, in the active place avoidance navigation task many of the cMPT-lesioned mice received more shocks during the training and probe trials than control animals (Fig. 6C). Moreover, in the passive place avoidance task, which requires cBF function[43], cMPT-lesioned mice displayed a significant memory deficit (Fig. 6D).

## Sleep deprivation does not result in cBF neuronal loss or Aβ accumulation

To determine whether the disrupted sleep or hypoxia induced by cMPT lesioning was responsible for causing cBF degeneration and Aβ accumulation, wildtype and aged APP/PS1 mice of both sex were subjected to sleep deprivation using a modified 'flowerpot' method (Supplementary Fig. 7A). It is difficult to disrupt NREM sleep, but this method results in a ~30% reduction in NREM sleep (mimicking the

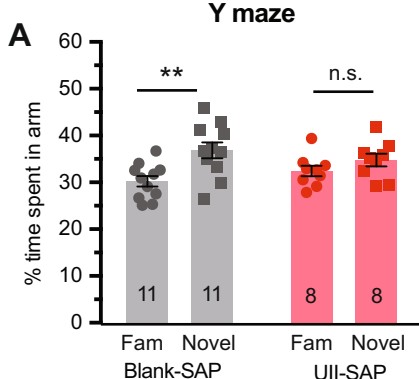

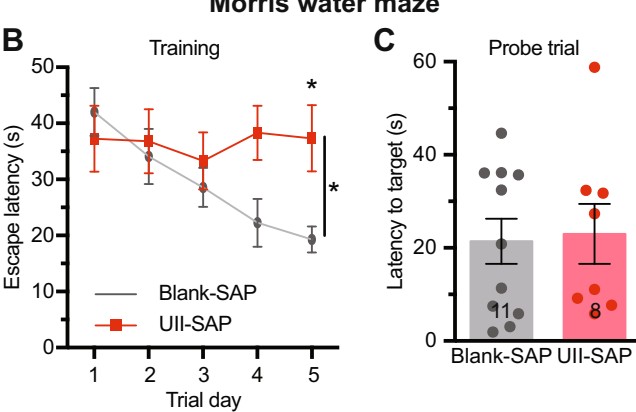

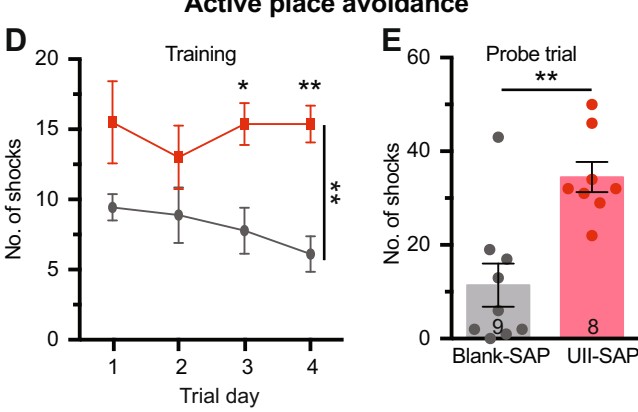

**Fig. 4 | cMPT lesion exacerbates cognitive impairment in APP/PS1 mice. A** The percentage time spent in the novel arm of the Y maze on test, compared to the familiar (Fam) arm. Blank-SAP-injected mice displayed a preference for the novel arm, whereas cMPT- lesioned mice had no preference ($P = 0.0040$, two-way ANOVA, Tukey's multiple comparison test; Blank-SAP: $P = 0.0072$, UII-SAP: $P = 0.6994$). **B** Escape latency in the training phase of the Morris water maze. cMPT-lesioned APP/PS1 mice spent significantly more time finding the escape platform on the last two training days ($P = 0.0150$, two-way ANOVA, Tukey's multiple comparison test; day 5: $P = 0.0376$). **C** Latency to reach the platform position in the probe test of the Morris water maze. There were no significant differences in escape latency between groups ($P = 0.8430$, Student's unpaired two-tailed t-test). **D** The number of shocks received by mice each training day of the active place avoidance test. cMPT-lesioned APP/PS1 mice received significantly more shocks on the last two training days ($P = 0.0044$, two-way ANOVA, Tukey's multiple comparison test; day 3, $P = 0.0159$; day 4, $P = 0.0022$). **E** The number of shocks the mice would have received in the probe trial of the active place avoidance test was significantly higher for cMPT-lesioned APP/PS1 mice than non-lesioned APP/PS1 mice ($P = 0.0012$, Student's unpaired two-tailed t-test). *$P < 0.05$; **$P < 0.01$; n.s., non-significant. Results are presented as mean ± s.e.m. Each data point represents an individual animal. Source data are provided in the Source Data file.

cMPT-lesioned phenotype) as well as almost completely eliminating REM sleep[44]. Cohorts of unlesioned mice were placed in sleep-deprivation cages for 20 h a day, returning to their home cage for 4 h a day during their sleep phase. The mice became visually sleepy over the course of the experiment, but maintained their body weight (Supplementary Fig. 7B, C). After 4 weeks of sleep deprivation, APP/PS1 animals displayed deficits in Y maze performance that were not exhibited by age-matched APP/PS1 mice housed in equivalent conditions except that their cages lacked water. However, contrary to expectation, latency to first sleep at the end of the paradigm was driven by age rather than cage type (two-way ANOVA, age: $P = 0.0113$ cage: $P = 0.2392$), with sleep deprivation causing long-lasting effects on the animals' sleep patterns. Sleep-deprived mice had normal diurnal sleep/wake rhythms ($P < 0.0001$ rhythmicity, $P = 0.676$ amplitude, $P = 0.943$ peak time) but a longer sleep latency and a reduction in total sleep time in the 3 days after the paradigm ended (Supplementary Fig. 7D–F). Nonetheless, the number of cBF neurons in the sleep-deprived mice was not significantly different from that in control mice (Fig. 7B). The levels of soluble and plaque Aβ (Fig. 7C, D) and degree of inflammation (Supplementary Fig. 7G–J) in the APP/PS1 mice were also unaltered by the sleep deprivation conditions. Taken together, our results indicate that SDB-induced hypoxia rather than sleep deprivation likely causes the cBF degeneration and exacerbates Aβ accumulation and inflammation. Nonetheless, the reduced inflammation observed in the control mice could be due to their lower level of stress compared to the sleep-deprived cohort, or as a result of their lower Aβ or acetylcholine levels.

### HIF1α mediates cBF death following cMPT lesioning

To test whether the cBF neurons were dying due to exposure to chronic hypoxic conditions, wildtype male mice were exposed to reduced (80%) oxygen conditions for 8 h a day during their sleep period for 4 weeks (Supplementary Fig. 8A). At the end of this period, the number of cBF neurons in mice subjected to daily chronic hypoxia was not significantly different from that of mice housed continuously in normoxia (Fig. 8A), indicating that chronic hypoxia was unable to trigger the same degenerative pathways that were induced by cMPT lesioning. In contrast, chronic intermittent hypoxia (with oxygenation of the air fluctuating from 0 to 100% every 90 s) has been reported to induce cBF neuronal degeneration[15].

To test whether fluctuating hypoxia/normoxia was the trigger for cell death, male cMPT-lesioned mice were treated for 4 weeks with the drug 2-methoxyestradiol (2ME2). This compound binds to and prevents the nuclear translocation of hypoxia-inducible factor 1 alpha (HIF1α), the expression of which is induced by low oxygen conditions and mediates the transcription of genes such as vascular endothelin growth factor to improve cerebral blood flow and oppose the toxicity of hypoxia. However, prolonged or persistent activation of HIF1α has also been linked to cell death mediated by pro-apoptotic Bcl-2 family members, e.g., via BNIP3, or increased expression of the reactive oxygen species-generating NOX proteins[45–47].

Two weeks after cMPT lesioning, wildtype mice were injected daily for either 3 (Supplementary Fig. 8B, C) or 4 weeks with 2ME2. As an indication that 2ME2 engaged its target, the percentage of cBF neurons with HIF1α present in the nucleus (indicative of its activation) in lesioned, untreated animals was significantly higher than that in both treated cMPT-lesioned mice, and sham-lesioned mice (Fig. 8B, C). The extent of cMPT neuron loss was equivalent between 2ME2-treated and untreated lesioned mice, and significantly reduced compared to the cMPT neuron number in sham-lesioned control animals (Fig. 8D; Supplementary Fig. 8B). However, the number of cBF neurons in the 2ME2-treated lesioned animals was significantly increased compared to that in untreated lesioned animals, and similar to that in sham-lesioned mice (Fig. 8E; Supplementary Fig. 8C). The 2ME2-treated

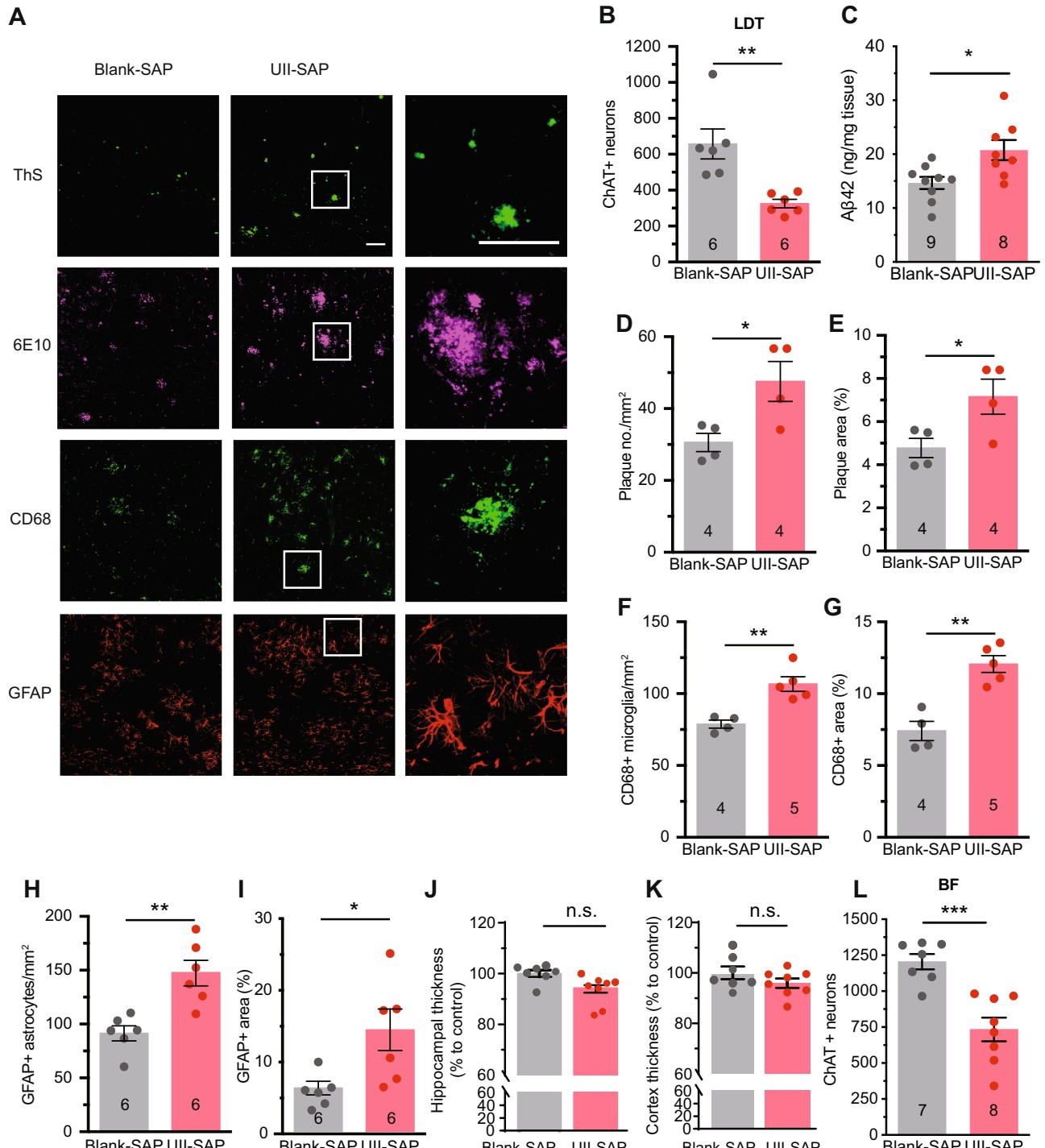

**Fig. 5 | cMPT lesion exacerbates major AD hallmarks of lesioned APP/PS1 mice.**
**A** Representative photomicrographs of sagittal brain sections of hippocampus and cortex of APP/PS1 mice treated with UII-saporin (6 animals) or control Blank-saporin (6 animals) and stained with thioflavin S (ThS) and 6E10 (both for Aβ plaques), CD68 or GFAP (for microglia and astrocytes). Scale bars = 100 µm. **B** The average number of ChAT-positive neurons in the LDT of UII-SAP -injected APP/PS1 mice was lower than that of APP/PS1 mice injected with Blank-SAP (*P* = 0.0033). **C** The amount of soluble Aβ in the hippocampal lysates of APP/PS1 mice treated with UII-SAP as measured by ELISA was higher than that of APP/PS1 mice injected with Blank-SAP (*P* = 0.0116). The density (**D**; *P* = 0.0320) and area (**E**; *P* = 0.0432) of thioflavin-S-positive Aβ plaque in neocortex of APP/PS1 mice treated with UII-SAP was higher than that of APP/PS1 mice injected with Blank- SAP. The density (**F**; *P* = 0.0028) and area (**G**; *P* = 0.0012) of CD68 immmunopositive staining in the neocortex of the APP/PS1 mice treated with UII-SAP was higher than that of APP/PS1 mice injected with Blank-SAP. Density (**H**; *P* = 0.0023) and area (**I**; *P* = 0.0241) of GFAP immunostaining in the neocortex of the mice treated with UII-SAP was higher than that of APP/PS1 mice injected with Blank-SAP. **J** The thickness of the CA1 pyramidal layer of UII-SAP-injected and Blank-SAP-injected APP/PS1 mice. **K** The thickness of the somatosensory cortex of UII-SAP-injected and Blank-SAP-injected APP/PS1 mice. **L** The average number of ChAT-positive basal forebrain (BF) neurons in APP/PS1 mice treated with UII-SAP was lower than that of APP/PS1 mice injected with Blank-SAP (*P* = 0.0038). *P < 0.05, **P < 0.01, ***P < 0.001 ****P < 0.0001 n.s.: non-significant. Student's unpaired two-tailed t-test for panels (**B**–**L**). Results are presented as mean ± s.e.m. Each data point represents an individual animal. Source data are provided in the Source Data file.

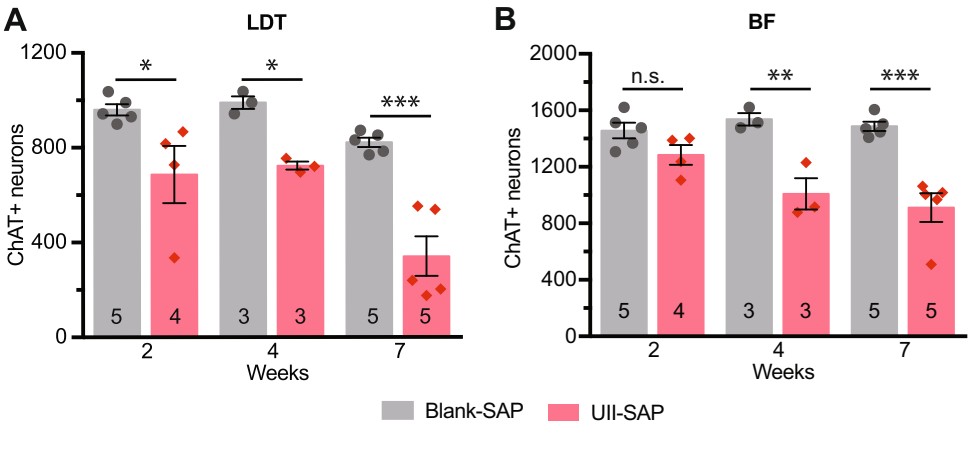

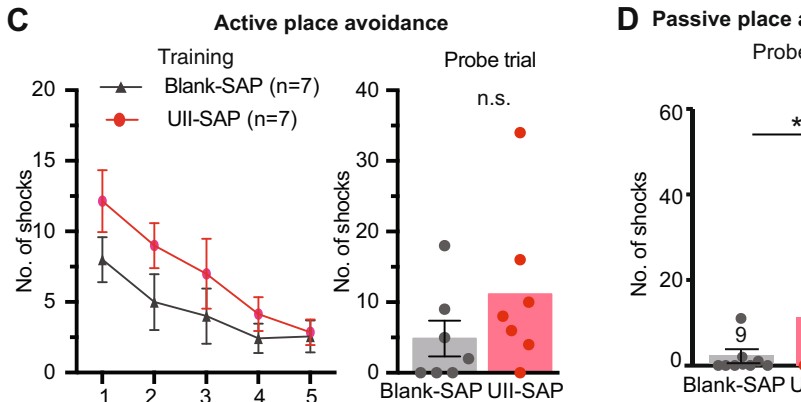

**Fig. 6 | cMPT lesions induce cBF neuronal degeneration and cognitive impairment. A** The number of ChAT-positive neurons in the LDT of C57Bl6 mice following injection of either UII-SAP or Blank-SAP 2-7 weeks earlier ($P < 0.0001$, two-way ANOVA, Bonferroni's multiple comparisons test, 2wk: $P = 0.0133$, 4wk: $P = 0.0248$, 7wk: $P < 0.0001$). **B** The number of ChAT-positive neurons in the basal forebrain (BF) of C57Bl6 mice following injection of either UII-SAP or Blank-SAP 2 to 7 weeks earlier. cBF neuron loss was subsequent to the loss in the LDT ($P < 0.0001$, two-way ANOVA, Bonferroni's multiple comparisons test, 2wk: $P = 0.3142$, 4wk: $P = 0.0010$, 7wk: $P < 0.0001$). **C** In the active place avoidance test, the number of shocks was not significantly different between conditions on any given training day

($P = 0.7580$, two-way ANOVA, Tukey's multiple comparison test) or in the probe test between the cMPT-lesioned mice and sham-lesioned mice ($P = 0.2279$, Student's unpaired one-tailed t-test), although a trend for poorer performance was seen. **D** The number of potential shocks recorded in the probe trial of the passive place avoidance test was significantly different between the UII-SAP and control blank-SAP groups. ($P = 0.04$, Student's unpaired one-tailed t-test) $^*P < 0.05$, $^{**}P < 0.01$, $^{***}P < 0.001$, $^{****}P < 0.0001$, n.s., non-significant. Results are presented as mean ± s.e.m. Each data point represents an individual animal. Source data are provided in the Source Data file.

lesioned animals also exhibited some improvement in memory in the cBF-dependent passive place avoidance task (Fig. 8F).

To determine whether the effect of HIF1α in cBF neurons was cell autonomous, we crossed conditional (floxed) HIF1α genetically modified mice with the mouse strain expressing cre recombinase from the ChAT gene locus (ChAT-IRES cre). Consistent with the 2ME2 results, the extent of cMPT neuron loss was significantly reduced in cMPT-lesioned cre-positive HIF1α$^{fl/wt}$ mice, compared to that in sham-lesioned cre-positive HIF1α$^{fl/wt}$ animals (not shown), whereas the number of cBF neurons in the lesioned ChAT-cre × HIF1α$^{fl/fl}$ animals was equivalent to that of sham-lesioned mice (Fig. 8G). These data indicate that the loss of cholinergic cBF neurons following cMPT lesioning is mediated by high endogenous HIF1α nuclear activity.

**Preventing hypoxia prevents AD features after cMPT lesion**

The standard treatment for OSA in human sufferers is the use of continuous positive airway pressure (CPAP) therapy during sleep, which physically maintains airway opening and thus prevents blood oxygen desaturation and arousal from sleep. We therefore asked whether treatment of the SDB mice during their sleep phase with a high-oxygen environment could protect cBF neurons from the cMPT lesion-induced cell death and/or Aβ accumulation. We again used the

high (40%) oxygen environment to restore the blood oxygen levels of cMPT-lesioned mice to >95% SpO$_2$ during their sleep phase (Fig. 3H). Starting 2 weeks after lesion surgery, aged APP/PS1 mice of both sexes were treated daily with high oxygen during their 12 h sleep period for 4 weeks. This treatment had no effect on the extent of the cMPT lesion (Fig. 9A). However, the number of cBF neurons in APP/PS1 (Fig. 9B; and wildtype Supplementary Fig. 3E) high oxygen-treated cMPT-lesioned animals was significantly higher than that of untreated lesioned mice, being similar to that of sham-lesioned animals in standard housing (Fig. 9B), thereby confirming that the cBF degeneration was a result of the SDB phenotype. Moreover, in the lesioned APP/PS1 cohort, the degree of amyloid pathology (Fig. 9C, E; Supplementary Fig. 9) and inflammation (Fig. 9F–H; Supplementary Fig. 9) were significantly reduced by the daily high-oxygen treatment.

## Discussion

Despite strong epidemiological links between SDB and the development of AD, the mechanisms underlying this increased risk are unclear. Here we developed a model of SDB in mice that replicates key features of the human condition without comorbid risk factors such as cardiovascular disease and diabetes. Using our model, we demonstrated that cBF neurodegeneration and cognitive impairment, two early

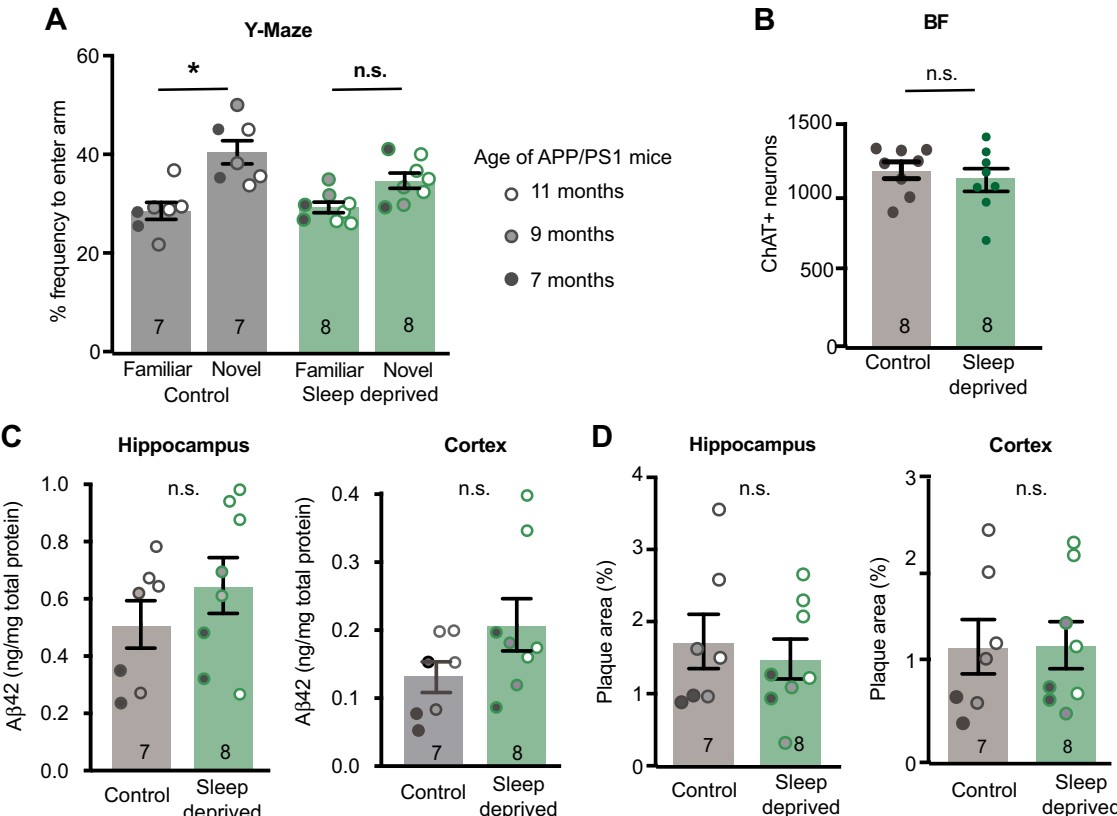

Fig. 7 | Sleep deprivation causes cognitive impairment but not AD pathology.
**A** The percentage of time spent in the novel arm of the Y maze on test, compared to the familiar arm. APP/PS1 mice displayed a preference for the novel arm, whereas sleep-deprived mice had no preference ($P = 0.0477$, control: $P = 0.0050$, Sleep-deprived: $P = 0.1733$). **B** The number of cBF neurons in C57Bl6 mice following sleep deprivation was equivalent to that of control mice ($P = 0.624$, Student's unpaired two-tailed t-test). **C** The amount of soluble Aβ in hippocampal (age: $P = 0.0602$, cage: $P = 0.774$, interaction: $P = 0.8960$) and cortical (age: $P = 0.0733$, cage: $P = 0.155$, interaction $P = 0.734$) lysates as measured by ELISA was not altered by sleep deprivation. **D** The area of thioflavin-S-positive Aβ plaque in the hippocampus (age: $P = 0.0330$, cage: $P = 0.9852$, interaction: $P = 0.6177$) and cortex (age: $P = 0.0128$, cage: $P = 0.410$, interaction $P = 0.666$) of APP/PS1 mice was affected by age but not by sleep deprivation. *$P < 0.05$, **$P < 0.01$, ***$P < 0.001$, ****$P < 0.0001$, n.s., non-significant. two-way ANOVA, Sidak's multiple comparisons for panels (**A**, **C**, and **D**). Results are presented as mean of pooled age groups ± s.e.m. Each data point represents an individual animal. Source data are provided in the Source Data file.

features of AD, are induced by intermittent hypoxia and that, in a genetically susceptible (familial) AD mouse, SDB exacerbates additional pathological features such as plaque load and inflammation. In contrast, although chronic sleep deprivation (for an equivalent period of time to that in which mice were subjected to SDB) impaired working memory of mice, it neither induced cBF neuronal dysfunction nor exacerbated Aβ accumulation. This suggests that fluctuating hypoxia is a major mechanism by which SDB, even in the absence of comorbidities, could promote AD in humans.

### Urotensin 2-saporin toxin lesion of the cMPT replicates key features of human SDB

Hallmarks of patients with untreated SDB include recurrent nocturnal arousals from sleep, caused by frequent episodes of pharyngeal obstruction. These cause hypopneas (≥4% decrease in arterial oxygen saturation[48]), and hypoxemia (<85% SpO₂ during sleep), resulting in wakefulness and frequent sleep interruptions. Our model displays hallmarks of a form of flow-limited SDB. cMPT-lesioned mice exhibited disrupted and variable breathing characteristics during sleep, specifically in EIP, PIF, EIP, and RT, and highly variable breathing frequency and tidal volume, which is consistent with upper airway resistance and subsequent recovery from hypoxia[29–31]. Furthermore, the mice exhibited hypoxemia, another feature of SDB with hypoxia, comparable to patients with moderate to severe oxygen-desaturation indices. Finally, the cMPT-lesioned mice had disrupted sleep patterns, with increased

transitions between REM and NREM sleep, reduced NREM sleep, and shorter sleep bouts that resulted in a sleep debt, the equivalent of human daytime sleepiness.

Although we cannot rule out the possibility that either the cMPT lesion or the resultant cBF neuronal degeneration contributed to the sleep phenotype independently of airway resistance, our data do not fit simply with such a conclusion. Stimulation (rather than inhibition) of neurons in the MPT suppresses NREM sleep phenotypes, and previous studies of lesions of the LDT report a minimal effect on REM sleep[21]. Similarly, inhibition of cMPT neurotransmission would be predicted to lengthen the time in NREM and shorten REM sleep bouts, the opposite to what we observed. Furthermore, we have previously reported that cBF-lesioned mice tend to take longer to rest during the light cycle, and are more active in the dark/active cycle[43]. In addition, cBF neuron lesions did not impair Y maze performance[43], whereas mice with the cMPT lesions or who have undergone sleep disruption have altered responses in this task, suggesting that mice with (only) cBF lesions are not sleep deprived. Nonetheless, control of breathing is regulated not only by upper-airway collapsibility, pharyngeal muscle responsiveness, and arousal thresholds, but also by 'loop gain'—the propensity of a system governed by feedback loops to develop unstable behavior, which is applicable to both central and obstructive forms of sleep apnea[49]. As both MPT and BF nuclei are heterogenous, selective cholinergic lesioning may have disrupted the balance of neurotransmission,

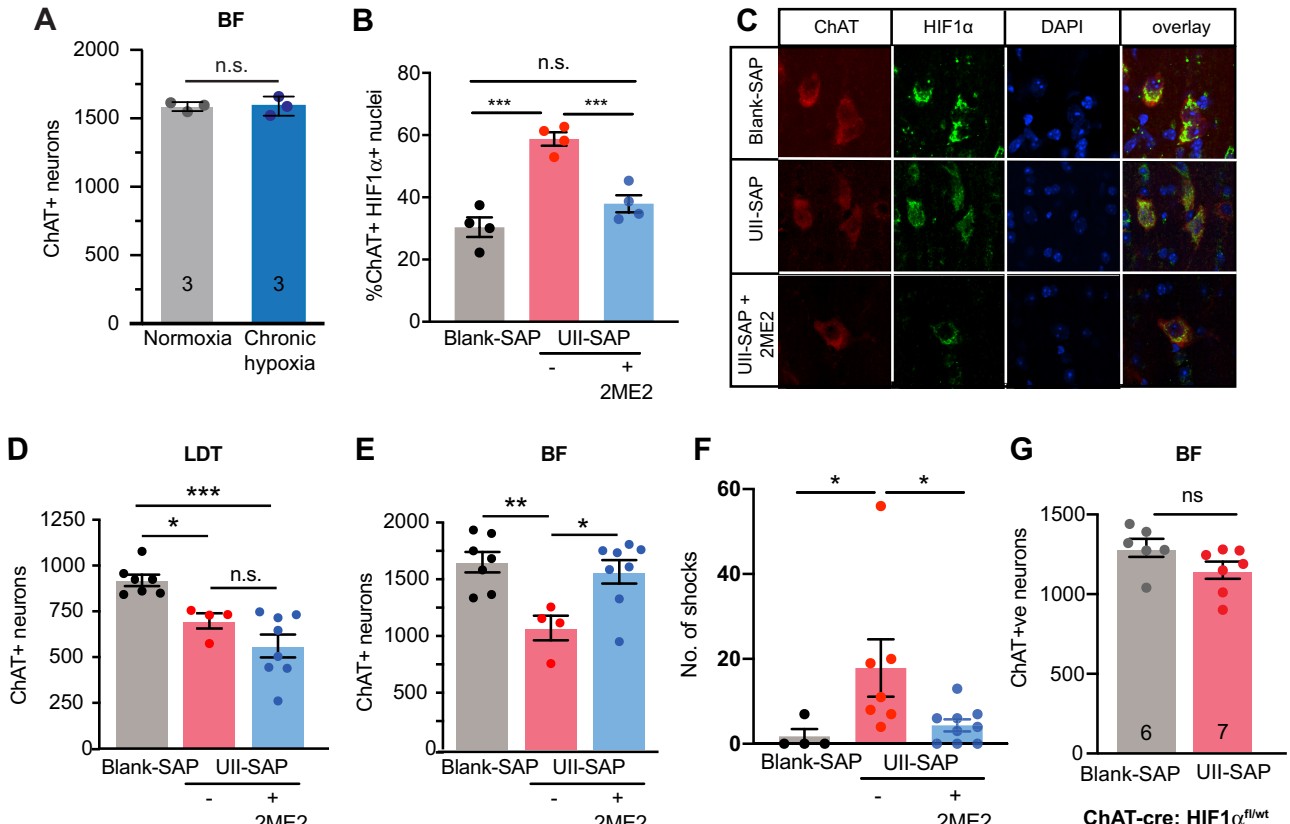

**Fig. 8 | cBF neuronal loss following cMPT lesion is induced by intermittent hypoxia. A** The number of cBF neurons in C67Bl6 mice following 4 weeks of daily sleep-time exposure to hypoxic conditions ($P = 0.3780$ unpaired two-tailed t-test).). **B** The percentage of cBF neurons in which HIF1α immunostaining was present in the nucleus in Blank-SAP (gray bar) or UII-SAP mice treated with daily 15 mg/kg 2ME2 (blue bar) or vehicle (red bar) for 3 weeks. (Blank-SAP vs. UII-SAP: \*\*\*$P = 0.0001$, Blank-SAP vs. 2ME2 treated: $P = 0.1766$, UII-SAP vs. 2ME2 treated: \*\*$P = 0.0011$). **C** Representative confocal images of basal forebrain sections from 4 animals immunostained for ChAT (red), HIF1α (green) and nuclei (DAPI; blue). The number of cLDT (**D**) and cBF (**E**) neurons in mice following injection of either Blank-SAP (gray bar) or UII-SAP and treated with daily 15 mg/kg 2ME2 (blue bar) or vehicle (red bar) for 3 weeks (cBF results: Blank-SAP vs. UII-SAP: $P = 0.0001$, Blank-SAP vs.

2ME2 treated: $P = 0.0020$, UII-SAP vs. 2ME2 treated: $P = 0.5401$). 2ME2 treatment protects cBF neurons from the effects of lesioning. **F** Performance of 2ME2-treated (blue bar) and untreated UII-SAP-lesioned (red bar) mice compared to Blank-SAP-lesioned mice (gray bar) in the test phase of the passive place avoidance task. (Blank-SAP vs. UII-SAP: $P = 0.0152$, Blank-SAP vs. 2ME2 treated: $P = 0.4014$, UII-SAP vs. 2ME2 treated: \*$P = 0.0112$). **G** The number of cBF neurons in UII-SAP-injected ChAT-cre HIF-1α$^{fl/wt}$ mice was not significantly different from that in Blank-SAP-injected ChAT-cre HIF-1α$^{fl/wt}$ mice ($P = 0.084$ unpaired two tailed t-test). \* $P < 0.05$, \*\*$P < 0.01$, \*\*\*$P < 0.001$ \*\*\*\*$P < 0.0001$ n.s., non-significant. one-way ANOVA Tukey's multiple comparison test for panels (**B**–**F**). Results are presented as mean ± s.e.m. Each data point represents an individual animal. Source data are provided in the Source Data file.

thereby increasing loop gain in response to airway restriction or natural apnea events, resulting in the observed variability and disruption to breathing during sleep. For example, GABAergic basal forebrain neurons stimulate arousal due to hypocapnia (high $CO_2$ levels in the blood)[50]. Our model will require further examination to determine whether the disordered breathing during sleep in the cMPT-lesioned mice accurately mimics a form of SDB in human (e.g. central or obstructive OSA)[51,52]. Analysis of breathing traces that can be manually assessed and scored (as occurs in human analyses) would provide a more thorough characterization of the changes in breathing induced by our experimental lesions. Nonetheless, our data, together with the finding that prevention of hypoxia during the sleep phase by keeping blood oxygen above 95% SpO2 averted a sleep debt, suggests that it is reasonable to conclude that the sleep disruption and hypoxemia we observed in cMPT-lesioned mice was triggered by limited air flow due to upper airway resistance which may or may not include hypopneas or apneas.

### SDB induces pathological hallmarks of AD

Untreated elderly patients with OSA exhibit cognitive decline at twice the rate, and develop AD at an earlier age, than the general population[7,8,13]. Similarly, in our study, induced SDB in older APP/PS

mice (already predisposed to overproduce Aβ peptides) resulted in more severe cognitive impairment than that observed in their age-matched littermates, and exacerbated many of the pathological features of AD, including induction of cBF degeneration, Aβ accumulation within the cortex and hippocampus, and increased inflammation represented by astro- and gliogenesis. However, even in the absence of a predisposition to AD pathology, SDB sequelae caused cBF degeneration and cognitive decline. cBF degeneration is associated with mild cognitive impairment in both humans[52] and rodents, particularly impairing higher order cognitive functions, such as allothetic and egocentric navigation, as measured in place avoidance tasks[34,43,53–55]. Although the hippocampal-dependent, spatial memory impairment of cMPT-lesioned APP/PS mice occurred concurrent with increased Aβ pathology (in the absence of hippocampal degeneration), the coincident cBF neuronal degeneration likely contributed to the behavioral phenotype[37]. These results indicate that even in the absence of other comorbidities, SDB can induce or enhance a range of phenotypes associated with AD development. However, as not everyone with SDB develops dementia, it will also be important to understand the characteristics of SDB that increased the development of AD phenotypes.

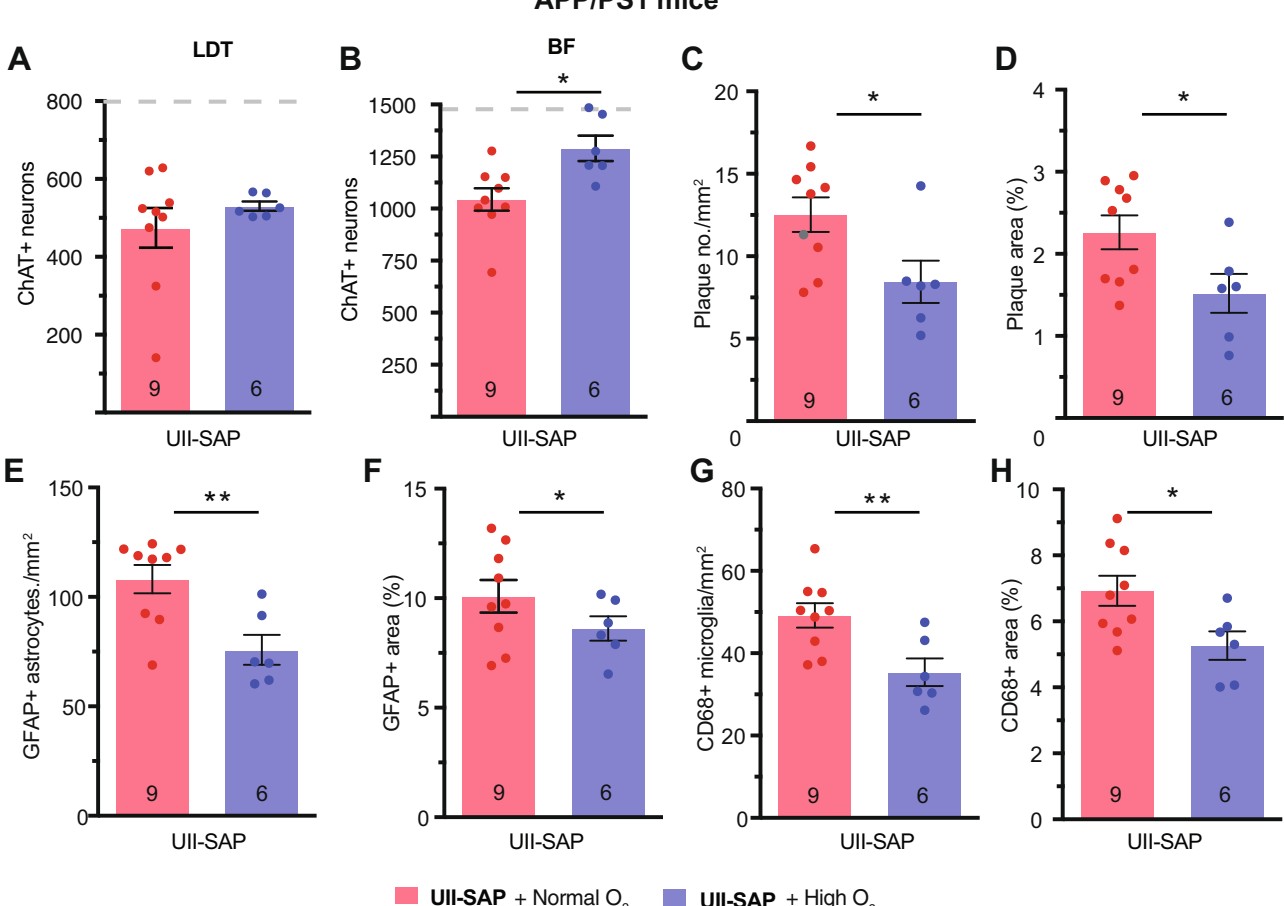

**Fig. 9 | High oxygen treatment protects from OSA-exacerbated AD phenotypes.** Number of cLDT (**A**) and cBF (**B**) neurons in APP/PS1 mice injected with UII-SAP and either the untreated (normoxia, normal oxygen) or treated daily with high oxygen. Number (**C**; $P = 0.0320$) and area (**D**; $P = 0.0432$) of thioflavin-S-positive Aβ plaques in the neocortex of APP/PS1 mice injected with UII-SAP and treated with normoxia or high oxygen. Density (**E**; $P = 0.0056$) and area (**F**; $P = 0.0432$) of GFAP-positive microglia in the neocortex of APP/PS1 mice injected with UII-SAP and treated with normoxia or high oxygen. Density (**G**; $P = 0.0098$) and area (**H**; $P = 0.0264$) of CD68-positive astrocytes in the neocortex of APP/PS1 mice injected with UII-SAP and treated with normoxia or high oxygen. $* P < 0.05$, $** P < 0.01$, n.s., non-significant. Student's unpaired two-tailed t-test for panels (**A**–**H**). Results are presented as mean ± s.e.m. Each data point represents an individual animal. Source data are provided in the Source Data file.

## Intermittent hypoxia underpins the risk of promoting AD features

Sleep deprivation is well known to cause reversible cognitive impairment in humans (e.g., ref. 56), and disruption of glymphatic system function, which can clear Aβ from interstitial fluid and is active during sleep, has been proposed as a mechanism by which poor sleep could be a risk for AD[57,58]. In our study, APP/PS mice subjected to sleep deprivation displayed impairment in Y maze performance. However, sleep deprivation alone was not sufficient to increase Aβ deposition or change Y maze performance in young wildtype animals. We cannot rule out a contribution to AD phenotypes, poor glymphatic clearance, or sleep disruption in our SDB model or in patients, particularly if sleep disruption causes persistent changes to sleep levels over a longer period. However, our results suggest that another feature of SDB, intermittent hypoxia, contributes more significantly to the progress of AD pathology than sleep disruption.

We determined that the death of cBF neurons was induced by cMPT lesioning but not by exposure to daily chronic hypoxia. Nonetheless, inhibition of HIF1α activity by 2ME2 or downregulation of HIF1α expression by heterozygous gene deletion in cBF neurons prevented their loss in cMPT-lesioned mice. These results indicate that cBF degeneration induced by SDB is dependent on cell-autonomous hypoxia/HIF1α-regulated pathways. Our conclusion is consistent with other reports in which rats exposed to intermittent hypoxia for 14 days (alternating 90 s epochs of 21 and 10% $O_2$ during sleep) developed deficits in water maze spatial memory performance and reductions in cBF neuron number that mirror the reduction in our SDB model[15]. Similarly, mice exposed to intermittent hypoxia during sleep (alternating 60 s epochs of 21 and 0% $O_2$ for 8 h) developed hippocampal synaptic changes and cognitive impairment in the Barnes maze spatial memory test[46], a memory test which are sensitive to cBF function[59]; these phenotypes were similarly prevented by heterozygous knockout of HIF1α[46].

Why intermittent and chronic hypoxia induce different cBF outcomes may be explained by cell culture experiments in which chronic hypoxia conditions cause HIF1α mRNA levels to be downregulated after several hours by negative feedback loops, whereas intermittent hypoxia does not activate the same transcriptional pathways, resulting in increased HIF1α mRNA levels[60]. The extent of HIF1α-induced downstream gene expression or repression also varies depending on exposure to chronic or intermittent hypoxia[60,61]. Counterintuitively, chronic hypoxia can also provide cellular protection[62]. We, therefore, propose that the fluctuations in hypoxemia observed in our SDB model are representative of intermittent hypoxia and reperfusion in the brain, which in turn is responsible for the observed cBF degeneration.

This conclusion is consistent with correlative findings from human populations. The cognitive decline in OSA is linearly correlated with hypoxemia[6,63], and a diagnosis of dementia in OSA patients has been associated with the extent of hypoxia rather than sleep disruption[5,13]. Cognitive impairment is also more common and more severe in OSA patients with hypoxemia[64,65].

Additional studies will be required to determine the physiologically relevant tissue-specific levels of hypoxia in our model, and the degree of oxygen volatility required to induce degenerative HIF1α-activated pathways in cBF neurons. Nonetheless, our work highlights fluctuations in hypoxia as the likely major factor in SDB that causes degenerative changes associated with AD.

### cBF degeneration is an early AD feature that can drive other AD features

Our finding that SDB exacerbates Aβ accumulation in genetically susceptible AD mice is consistent with a previous report that exposure of mice to chronic intermittent hypoxia increases Aβ42 levels in a triple AD model mouse[14]. Similarly, higher brain Aβ[66], or decreased Aβ and increased phosphorylated tau in the cerebrospinal fluid[67,68], has been associated with a higher oxygen desaturation in cognitively impaired OSA subjects. One explanation for these observations could be that hypoxia and HIF1α-mediated transcription can directly up-regulate the expression of β-secretase[69], while HIF1α can operate as a regulatory subunit of γ-secretase during hypoxia[70], both of which could increase Aβ production via the amyloidogenic pathway. However, exacerbation of Aβ accumulation in mice that are already overproducing the peptide could also have occurred through an indirect mechanism, specifically cBF degeneration. Although in this study, we did not determine whether cBF degeneration occurred in parallel with, or exacerbated the other AD pathology, previous research strongly implicates the latter.

cBF neuronal lesioning in a range of mouse AD models exacerbates many features of AD, including promoting cognitive impairment, and increasing Aβ accumulation, plaque load, and inflammation (as observed herein), as well as causing tau hyperphosphorylation, hippocampal degeneration, and reduced neurotrophin expression[40–42,71,72]. On the other hand, AD model rodents with increased cBF connectivity[73,74] or those which are treated with muscarinic receptor agonizts[75] display cognitive improvements and significantly reduced age-dependent Aβ accumulation and inflammation compared to control rodents. Cholinergic activation of muscarinic receptors also stimulates non-amyloidogenic APP processing[76], thereby regulating the levels of Aβ in the interstitial fluid[77]. Similarly, in human subjects, reduced basal forebrain volume (as measured by structural MRI, and indicative of cholinergic degeneration) predicts Aβ burden[33,35,37,55,78] and is evident in mild cognitive impairment and idiopathic AD cohorts[37,53,55], as well as in individuals who are subsequently diagnosed with cognitive impairment[33,34,53]. Furthermore, basal forebrain degeneration, together with the accumulation of Aβ, precedes and causes entorhinal cortical degeneration in humans[79]. Therefore, it is probable that the hypoxia-induced death of cBF neurons directly contributed to the other AD phenotypes in our SDB model, and may represent a decisive factor in the etiology of idiopathic AD. Determining the extent of cBF degeneration in people suffering from intermittent hypoxia as a result of SDB would test this interpretation.

Finally, the effect of preventing hypoxemia and sleep disruption in our SDB mice during their daily sleep period, protected against the development of the phenotypes associated with AD. The direct equivalent treatment for humans would be sleeping in a hyperbaric chamber, but high levels of oxygen is not necessarily healthy (e.g. it can result in oxidative stress). However, emerging studies indicate that CPAP treatment of people with OSA to reduce their hypoxia may reduce neurodegeneration[80] and delay the progression of cognitive impairment[81], particularly that relating to attention and executive function in which cBF neuronal function is strongly involved[5]. The implication of these studies, together with our findings that the prevention of hypoxemia during the sleep period in cMPT-lesioned mice reduced cBF degeneration and Aβ accumulation, is that early diagnosis and treatment of OSA with CPAP may significantly decrease the longitudinal risk of developing irreversible cognitive impairment. Our study therefore provides a rational impetus to longitudinally study CPAP as a prophylactic treatment to reduce the incidence of AD in people with SDB[82].

In summary, our results suggest that SDB, even in the absence of other comorbidities, is a risk factor for the development of AD. Specifically our work highlights hypoxia as the decisive feature, as it induces cBF neuronal degeneration and cognitive impairment, with a coincident or subsequent increase in the levels of Aβ when linked to a genetic risk for Aβ accumulation (potentially including an ApoE4 genotype)[5,68]. Although treating the hypoxemia in our SDB model or giving CPAP in humans reduces respiratory disturbance during sleep, and improves working memory, some cognitive aspects such as complex attention and executive function remain impaired[6], daytime sleepiness can continue, and Aβ load can remain unchanged[10,11], all of which might reflect irreversible cBF neuronal loss that predates the commencement of CPAP treatment. Therefore, targeting cholinergic function (through cholinesterase inhibition or muscarinic agonizts)[33,83] or specific HIF1α pathways (2ME2 is a current treatment for tumor angiogenesis) represents a candidate strategy for preventing AD in the SDB 'at risk' population—those with intermittent hypoxia.

## Methods

### Animals
C57BL/6, APP/PS1 (JAX #34832), and ChAT-IRES-cre (JAX #006410) × Hif1α[fl/fl] (#007561; sourced from the Victor Chang Cardiac Research Institute and outcrossed to C57BL/6 prior to heterozygous breeding for experimental use) mice were maintained on a 12 h light/dark cycle in Optimouse cages with *ad libitum* access to water and food (Specialty Foods, SF00-100, and SF00-105; the latter for animals aged 1 year and older). Littermates of the same sex were randomly assigned to experimental groups and used at 8–12 weeks or 8 months (for APP/PS1 mice) of age unless stated otherwise; both male and female APP/PS1 mice were used and results were combined as no differences were found between sexes. In other experiments, males and females were analyzed separately unless otherwise stated, or all male cohorts were used due to their more severe SDB phenotype. All procedures were approved by the University of Queensland Animal Ethics Committee (approval numbers 2012/AE000407, 2012/AE000029, 2015/AE000534, 2015/AE000084, 2018/AE000566, and 2018/AE000135) and conducted in accordance with the Australian Code of Practice for the Care and Use of Animals for Scientific Purposes.

### Surgery
Standard surgical procedures were followed for stereotaxic injection[41,42]. Animals were anesthetized by intraperitoneal (i.p.) injection of ketamine (100 mg/kg) and the muscle relaxant xylazine (10 mg/kg; lesion surgery) or 2–3% isoflurane in 100% oxygen (radiotelemetry implants). Mice received post-operative 0.1 mg/ml Meloxicam (Troy Laboratories) subcutaneously for three days. To induce cholinergic mesopontine tegmentum (cMPT) lesions, bilateral injections of urotensin II-saporin (UII- SAP; 0.07 µg/µl per site—unless stated otherwise; generous gift from Advanced Targeting Systems) in the laterodorsal tegmental nucleus (LDT; AP, −3.0 mm; ML, ±0.5 mm; DV, −4.0 mm from Bregma, with the angle 34° backward) were made using a calibrated glass micropipette through a Picospritzer® II (Parker Hannifin), with the same molar mass of blank-saporin (Blank-SAP) or IgG-saporin (IgG-SAP) as control.

To measure sleep states, we used HD-X02 radiotelemetry implants from Data Science International to measure electroencephalography (EEG), electromyography (EMG), and activity rhythms (for details see refs. [84], [85]). The EEG electrodes were placed epidurally above the motor cortex (AP+1, ML+1 mm left from bregma) and the visual cortex (AP−3, ML+3 mm right from bregma). EEG electrodes were fixed with glass ionomer cement (Kent Dental). EMG electrodes were anchored in the trapezius muscle. The scalp was closed using nonabsorbable 6-0 suture material (Ethicon, USA). Approximately 7% of Blank Sap-treated mice and 12% of UII-SAP-treated mice were excluded due to health reasons related to the surgical procedure. Where UII-SAP-treated mice did not show >10% cell loss formt he mean of the control cMPT cell number they were excluded from other lesion group analyses.

### Phenotypic analysis

Behavioral experiments were always performed in the light cycle. Although this was during the animals' sleep cycle, we reasoned, that this timing permitted sleep-deprived animals to have a chance to sleep prior to the testing. Some cohorts of mice were subject to multiple behavioral tests.

**Sleep assessments.** Two weeks after electrode placement surgery, EEG and EMG were recorded for up to 48 h at a sampling frequency of 500 Hz and captured by the Ponemah 6.4 software (DSI, USA). EEG and EMG recordings were scored in 4 s epochs by the automated rodent sleep-scoring software, NeuroScore 3.0 as wake, REM or non REM (NREM) sleep. Sleep scoring was also supported through video surveillance and an expert assessor. REM sleep bouts were only considered if they were ≥ 1 s long. Two control mice in which overt brain damage was observed due to electrode placement were excluded from the analysis.

A separate cohort of mice was assessed over 72 h for sleep/wake parameters using a metabolic phenocage system (Phenomaster; TSE Systems) with ActiMot light sensors of movement and pre-programmed control of light cycles and light brightness. Mice were considered to be asleep if they had ≥10 min of no movement.

**Whole-body plethysmography.** Following EEG, mice were assessed using whole body plethysmography[86] for up to 4 h to record respiration while simultaneously recording EEG in otherwise unrestrained, freely moving mice. The mice were placed in a plethysmograph chamber (Buxco FinePointe Series WBP, DSI) continuously filled with fresh air at room temperature. This approach provides an indirect measure of tidal volume, which is directly proportional to the cyclic chamber pressure signal produced during respiration in a sealed chamber. The mice were first allowed 30 min to acclimatize to the chamber, after which the recording session proceeded for 3 h. Breathing measures recorded every 2 s were time-linked to the EEG-predicted sleep state. Only data points for which the sleep/wake state was identified were included in the analysis, with no other exclusion criteria being applied post data collection. However, mice showing >3 standard deviations from the mean in EEG were excluded from analysis. One lesioned mouse with a 10 fold reduced breathing frequency during sleep (compared to all other mice) was excluded from analysis. For each mouse, the values for each breathing measure during sleep (REM+NREM) and the wake period were averaged. Data was analyzed by two way ANOVA (sex, lesion) pairing the sleep and wake state measured for each mouse. The variability of each parameter for each mouse was also calculated using Levene's test of equality of variances in which, for each mouse, that average value for each parameter was subtracted from all values and the resulting data compared between cMPT-lesioned or control groups by one way ANOVA.

**Arterial oxygen saturation.** The arterial oxygen saturation was recorded on unrestrained awake mice either in room air or in the environmental chamber using a MouseOx Plus oximeter (Starr Life Sciences) in accordance with the manufacturer's instructions. Data were collected for at least 30 min and only used when no error code was given. Oxygen saturation levels were then calculated as an average per mouse, and per experimental condition.

**Open field.** The open field test was performed in a square white Plexiglas box (30 cm × 30 cm) for 30 min. The chamber was divided into a central field (center, 15 cm × 15 cm) and an outer field (border). The mouse's movement was recorded using a video camera, and analyzed using the EthoVision XT video tracking system v14/15 (Noldus Information Technology). Total track length was calculated using the center of the animal's body as the reference point.

**Y maze.** The Y-maze was composed of three equally spaced arms (120° apart, 35 cm long, and 10 cm wide) made of Perspex. During training, the mice were placed in one of the arms (the start arm) with access to only one other arm for a period of 15 min. One day after the training, mice were allowed to explore in the Y-maze for 10 min with access to all three arms. Animals were tracked using EthoVision XT software, and the total time and the percent time spent in each arm were analyzed.

**Novel object recognition.** The task procedure consisted of three phases: habituation, familiarization, and the test phase. In the habituation phase, the mice were allowed to freely explore the open-field arena (40 cm × 40 cm) for 5 min. During the familiarization phase, they were placed in the arena containing two identical sample objects, for 5 min, where both objects were located in opposite and symmetrical corners of the arena. After a 1 h retention interval, the mice were returned to the arena with two objects, one identical to the sample and the other novel, and were allowed to explore again for 5 min. Animals were tracked using EthoVision XT software and the percent time spent exploring the objects were analyzed.

**Active place avoidance.** The apparatus (Bio-Signal Group) consisted of an elevated arena with a grid floor fenced with a transparent circular boundary, located in a room with visual cues on the walls[87]. The arena rotated clockwise (1 rpm) and an electric shock could be delivered through the grid floor. On the day after habituation, mice underwent daily training sessions for 4 or 5 days, lasting for 10 min per day in which the mouse was trained to avoid a 60° shock zone, entrance to which led to the delivery of a brief foot shock (500 ms, 0.5 mA). To determine their memory of the shock zone, 24 h after the last training session, the mice were allowed to explore the arena for 10 min without shocks being applied. The position of the animal in the arena was tracked using an overhead camera linked to Tracker software v2.1 (Bio-Signal Group).

**Passive place avoidance.** The passive place avoidance behavioral paradigm was used to test basal forebrain-dependent idiothetic navigation[42,87]. This navigation task used the same apparatus as the active place avoidance task. However, all external cues were eliminated to minimize allothetic navigation, the circular boundary was an opaque gray, and the arena remained stationary throughout the experiment. The mice went through a 5 min habituation session and then a 10 min training phase with a 1 h interval between them. Twenty-four hours after the training session, the mice were allowed to explore the arena for 5 min without shocks being applied[42].

**Morris water maze.** Mice were trained to escape the maze by finding a platform submerged 1.5 cm beneath the surface of the water and invisible to the mice while swimming. Three trials were performed per day, or 60 s each with 30 min between trials. During each trial, mice

were placed into the tank at one of three designated starting points in a pseudorandom order. If the mice failed to find the platform within 60 s, they were manually guided to it and allowed to remain there for at least 20 s. Mice were trained for 5 days as needed to reach the training criterion of 15 s (escape latency). The probe trial occurred 24 h after the last training session and consisted of a 60 s trial without the platform. Performance was measured following tracking with the EthoVision XT video tracking system.

**Sleep deprivation.** To induce sleep deprivation, up to 8 mice were placed in a water-filled cage (55 cm × 25 cm) that contained 9 cylinders (3 cm diameter) with 1 cm above the water surface (Supplementary Fig. 7A). The sleep deprivation cages and control cages, which also contained cylinders but no water inside, were held within a climate chamber on a 12:12 h light:dark cycle at 27 °C and 25% relative humidity. Mice were placed in the sleep deprivation cages for 20 h a day and in their home cages (4 to a cage), also within the climate cabinet, for 4 h per day during their light cycle. Mice were monitored by video and weighed daily.

## 2ME2 treatments
Two weeks post-surgery, the cMPT-lesioned mice were treated with either a HIF1α inhibitor 2- methoxyestradiol (2ME2, Sigma-Aldrich, M6383), or vehicle control (saline with 0.5% DMSO). The 2ME2 was dissolved in saline at a concentration of 0.5 mg/ml with 0.5% DMSO. Mice received daily i.p. injections of either 2ME2 (15 mg/kg body weight) or vehicle for 3 or 4 weeks. Mice were weighed every third day to detect any body weight changes and possible overt toxicity of long-term 2ME2 administration (no changes were observed).

## Immunohistochemistry
For immunofluorescence labeling, free-floating sections were probed using the following primary antibodies: goat anti-ChAT (1:400, AB144P, Millipore), mouse anti-parvalbumin (1:1000, MAB1572, Millipore), rabbit anti-calbindin (1:2000, CB38, Swant), mouse anti-Aβ (6E10, 1:500, Sig-39320, Convance), rabbit anti-GFAP (1:500, Z0334, Dako), rat anti-CD68 (FA-11, 1:500, MCA1957, AbD Serotec), and rabbit anti-HIF1α (1:200, NB100-479, Novus Biologicals) followed by the appropriate secondary antibody (1:1000, Life Technologies) or incubated with thioflavin S (0.1% in water, T1892, Sigma-Aldrich). Sections were mounted onto slides and coverslipped using fluorescence mounting medium (Dako).

For DAB (3,3'-diaminobenzidine) staining of cholinergic axons in the cortex, the biotinylated donkey secondary antibodies (1:1000, Jackson ImmunoResearch Laboratories) and ABC reagent (Vector Laboratories) were applied following the primary antibody incubation and followed by the nickel-intensified DAB reaction. Brain slices were mounted on slides and coverslipped with DPX mounting medium (Sigma-Aldrich).

## ELISA methods
For biochemical analyses, mice were perfused with PBS, after which the cortex and hippocampus were dissected, weighed, and snap-frozen with liquid nitrogen. The protein was extracted by adding ice-cold 10 v/w RIPA buffer (250 mM NaCl, 1% NP-40, 0.5% sodium deoxycholate, 0.1% SDS, 50 mM Tris HCl, pH 8.0, containing Roche cOmplete™ protease inhibitor cocktail and PhosSTOP phosphatase inhibitor cocktail and 1 mM AEBSF) to the tissue before homogenization using Bullet Blender Storm 24 (Next Advance).

The level of Aβ in the tissue was assessed by ELISA as per the manufacturer's instructions (KHB3442, Invitrogen). The resulting measurements were normalized for tissue weight.

## Image analysis and histological quantification
Imaging was performed using either an upright fluorescence microscope (Axio Imager, Zeiss Zen; 2012, Carl Zeiss) Slidebook v6.0 (3I Inc), Nikon NIS, and VSlide Scanner (Metafer, Metasystems), or a Diskovery spinning disk confocal microscope (Andor, Oxford Instruments). All measurements and analyses were performed using Imaris 9.2.1 software (Bitplane) or Image J v1.45 (NIH). MPT and basal forebrain cell counts were obtained from every 3rd section of a 3 series of the relevant anatomical region as defined by the Mouse Brain Atlas. The total cell count per mouse was summed and graphed. UII-SAP-injected mice not displaying a loss of cMPT neurons of >10% from the mean of controls were excluded from further analysis. Axonal density analyses of cBF neurons was undertaken using ImageJ. For each animal, 5 somatosensory cortex-containing brain slices were selected at random and the level of ChAT immunostaining was analyzed by measuring the gray value along one line per cortical layer. These values were then pooled, averaged, and inversed to obtain the average axon density for each animal (Boskovic et al.[73]).

## Statistical analysis
Statistical analysis was performed using GraphPad Prism software (versions 7–9), accounting for appropriate distribution and variance to ensure proper statistical parameters were applied. Prism files containing source data and statistical outcomes (including confidence intervals) are available in the Prism files in the Source data. Levene tests were performed using the program R. Graphed values are expressed as the mean ± s.e.m (except where stated as s.d) with significance determined at $P < 0.05$. Data from different cohorts of mice that had undergone the same procedures were combined for analyses.

## Reporting summary
Further information on research design is available in the Nature Research Reporting Summary linked to this article.

## Data availability
Source data are provided with this paper as a Source Data file. This together with raw and processed data generated in this study, including statistical calculations, and have been deposited at: https://doi.org/10.48610/81462a6. Any raw data files not deposited are available from the corresponding author. Source data are provided with this paper.

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

## Acknowledgements

This work was supported by the National Health and Medical Research Council of Australia (GNT1049236 to E.J.C. and M.C.B.; GNT 1162505 to E.J.C.; GNT1186943 to O.R.), Dementia Australia Research Foundation, the Brain Foundation, the Mason Foundation and the Clem Jones Center for Ageing Dementia Research. We thank Advance Targeting Systems for their generous supply of UII-SAP and Blank-SAP. Work was performed in the School of Biomedical Sciences' and Queensland Brain Institute's Advanced Microscopy Facilities with support from Shaun Walters and Luke Hammond respectively, and the Queensland Brain Institute's Animal Behavioral Facility with support from Daniel Blackmore, and the School of Biomedical Sciences Integrated Physiology Facility with support from Stuart Mazzone and Melanie Flint. We thank Queensland Brain Institute's

Animal Facility staff, particularly Liesel MacDonald, as well as Michael Lardelli (The University of Adelaide, Australia) for helpful suggestions regarding hypoxia, Bree Rumballe for excellent laboratory mangement and discussion, and Rowan Tweedale for manuscript editing.

## Author contributions

L.Q., O.R., M.C.B., and E.J.C. designed the experiments, L.Q., O.R., L.K., M.R.M., N.G., K.S., N.M., P.R.K., and A.S. performed experiments, L.Q., L.Q., O.R., L.K., M.R.M., N.G., K.S., N.M., M.W.D., M.C.B., and E.J.C. analyzed data, all authors discussed the work, and wrote or commented on the manuscript.

## Competing interests
The authors report no competing interests.
