## [Peer Review File · Nature Communications]

Cholinergic basal forebrain degeneration due to sleep-disordered breathing exacerbates pathology in a mouse model of Alzheimer's diseaseREVIEWER COMMENTS

Reviewer #1 (Remarks to the Author):

This interesting study aimed to explore the impact of obstructive sleep apnea (OSA) combined with death of cholinergic neurons in basal brain nuclei on the progress of Alzheimer's disease (AD) in engineered model mice carrying a mutated human beta amyloid gene and subjected to Saporin injection for destroying the p75 receptor and exacerbating the deterioration of deep cholinergic nuclei. The topic is of great interest to many, and previous reports were less authoritative than hoped since the employed model systems were not optimal. In the current manuscript, the authors describe a novel OSA model based on much effort and care that was established as precisely as possible, which is a major strength point of this study. While the induced disease in these mice involves both a human beta amyloid transgene and impaired sleep, and although the injected saporin affects the murine brain, avoiding the OSA diminishes the exacerbated disease symptoms in tested mice. This provides experimental support to the authors' working hypothesis and strengthens their claims for the relevance of using air pressure therapy for delaying the progress of familial AD. However, this study also entails several key points which raise key questions and call for additional experimental work, as is detailed below.

Major comments

1. While the abstract describes 'oxygen-sensitive p75 receptors' as causally involved in the OSA-induced exacerbation of AD, the basal nuclei p75 receptors in the mesopontine nuclei of the currently studied mice were destroyed by saporin injection; although suppressing the OSA showed its causal relevance in the observed symptoms, the saporin poisoning raises a question regarding the role of p75 receptors in basal brain nuclei in the impact of OSA and must be declared in the abstract and referred to in the results and Discussion sections in larger detail.
2. The study involved both male and female mice and does not distinguish between sexes. However, ample reports demonstrate that women are more readily susceptible to develop AD, and that they deteriorate faster when the disease progresses even when their longer life expectancy is factored in. However, the underlying mechanisms are still unknown, and the current study raises the question of OSA might be causally involved. Therefore, performing the reported tests separately in male and female mice, and comparing their susceptibility to OSA based on experimental evidence would be most valuable.
3. Considering the topic of this study and the recently proved importance of specific muscarinic receptors for REM sleep, quantifying the levels of such receptors and/or their coding transcripts and citing the corresponding Ueda references may offer a molecular link and a more precise measure of the affected processes.
4. Apart from the deep nuclei, AD involves escalated thinning of the cortical layer where much of the cholinergic input is provided by VIP-ACh interneurons. Quantifying these neurons as well could add another causal element to the study of OSA-induced AD and should be tested.
5. The navigation learning capacities of tested mice appear to be only partially damaged, possibly indicating that OSA affects hippocampal functioning less dramatically than its impact on other brain regions. In this context, measuring changes in hippocampal volume in the affected mice may be of interest.
6. Given the importance of neuroinflammation for the progress of AD, quantifying the neuroinflammation extent should be presented in the main body of this manuscript and involve in-depth statistics, again separately in male and female animals.
7. To more substantially prove the causal involvement of HIFa in the damaged CMPT neurons, one should test the impact of these neurons on the OSA-induced exacerbation of AD progression with and without interruption of HIFa expression in CMPT neurons.
8. Considering the recent publications reporting an increased risk of dementia in aged patients under chronic anti-cholinergic medications, it is logical to quantify the cholinergic tone in these mice, possibly by measuring cholinesterase activities in the circulation.
9. Apart from OSA patients, this study may infer added value for treatment with high oxygen levels as delaying AD symptoms severity. This needs to be discussed.

Minor comments

1. The cited Hardy and Selkoe review in the introduction is almost 20 years old; wasn't there anything fresher which is worth citing since 2002?!
2. The experimental use of whole body plethymography has been pivotal for reaching the current conclusions, and may be mentioned in the title.
3. To ensure maximal impact of the observed phenomena, the data availability should be open, even if in a local website; and not dependent on personal requests from the corresponding author.
4. Employing commercially available mice with fluorescently labeled ChAT expressing neurons may offer a better follow-up of the loss of such cells in the mesopontine nuclei.
5. Adding more information on the OSA parameters in human patients may provide a valuable comparison to the current study and increase its biomedical value.
6. The discussion should address the different impact of chronic sleep loss vs OSA in a more in-depth manner.
7. The added value compared to the Shiota et al., 2013 article should be highlighted.
8. Fig 2H presents data of individual mice which should be replaced by average group values in male and female animals.
9. Fig 1, 4, 7 should all include size bars in all pictures, with enlarged cells shown for better identifying the presented differences. Correction of three dimensional errors is recommended.
10. Supplementary Fig 5 shows reduced astrocytosis in mice under OSA, which is confusing as it might reflect lower stress levels. This should be explained.

Reviewer #2 (Remarks to the Author):

Qian and colleagues conduct a very interesting study using a mouse model to establish a link between sleep apnea and Alzheimer's pathology. Two factors mediate the pathological changes: oxygen sensitive p75 neurotrophin receptor and hypoxia inducible factor 1 alpha. Sleep apnea was induced by producing cMPT neuron lesions, using UII-SAP injected directly into the LDT and intraventricularly. Oxygen therapy prevented the formation of the Alzheimer's pathology. Overall, the studies seem well conducted and provide good evidence for a physiological mechanism through which sleep apnea can lead to Alzheimer's disease.

My primary concern about the study is the statistical analyses. The statistical analysis section offers no discussion of how the data are analyzed. There is no mention of statistical tests or methods used to compute confidence intervals. No confidence intervals are reported. The statistical analyses entail primarily comparisons of the differences in two independent samples, suggesting two-sample, t-tests, z-tests and/or Mann-Whitney non-parametric test. Most of the samples are small. The largest samples have 15 animals per group. Reporting p-values is a very weak approach for making inferences because all they say, when results are found to be significant, is that the null hypothesis is improbable. Two things should be done here. The authors should make their comparisons by constructing confidence intervals for the median differences between the groups. The inference is made, i.e. the two groups would be considered different, if the 95% or 99% confidence intervals do not contain zero. The important added advantage of the confidence intervals is that they make it possible to measure directly the magnitude of the differences between the two groups and the uncertainty in those differences. To really explore the variability in the data these confidence intervals should be computed by non-parametric bootstrap methods. If analyses are redone in this manner, I suspect that several of the results which are reported here as significant by the current analyses will no longer be significant. The authors should then reassess the validity of their inferences. It may be necessary to change the conclusions and or conduct more studies.

The statistical tests are mentioned in the figure legends. They should be discussed in the Methods section of the paper.

Reviewer #3 (Remarks to the Author):

This is a very interesting, often fascinating, study in a topical and important area of investigation. Several approaches taken employ state-of-the-art experimental techniques. Several data sets are very convincing and undoubtedly support the conclusions drawn. However, there would also appear to be some short-comings and conclusions that are over-stated and/or confusing (at least to me).

I commend the authors on a series of experiments that are elegant and carefully conducted. The data are reported in transparent fashion. There is a coherent basis to the sequence of experiments performed, culminating in some fascinating observations that provide mechanistic insight to p75NTR-HIF-1 α -dependent neuronal cell loss implicated in exacerbation of AD pathology.

I was less convinced by some of the data and study design in early experiments, that make it difficult to determine if disrupted sleep causes disrupted breathing or vice versa in this new model. In my view, there are some problems here that ought to be acknowledged in the manuscript. There are other experimental approaches that would add considerable rigor. However, the studies illustrating convincingly that 2ME-2 treatment, or p75NTR selective knockout, or oxygen therapy each prevent cBF neuronal cell loss and impaired cBF-dependent behaviours (or at least some of them) are impressive.

A slight worry is that the authors did "everything" but administer intermittent hypoxia, which it is argued, is the driving pathological feature (and undoubtedly is relevant to OSA). Perhaps it is because this has already been done (Row et al. 2007) showing CIH-induced cBF neuronal cell loss and cognitive impairment. This old paper rather steals from the major conclusion of the present study in that the argument has been made already. The present study extends the argument to reveal a potentially causal signalling pathway and possible druggable target. Since the authors are well placed to perform studies of intermittent hypoxia (given the study of daily sustained hypoxia), the final piece in the puzzle i.e. showing that IH drives pathology via p75NTR-HIF-1 α is noticeably absent. I highlight this because although the argument can be readily made it is not entirely conclusive from the collective data in the current study.

Of course, I recognise that this in turn might provoke interest in the efficacy of 2ME-2 or disruption to p75NTR in ameliorating IH-induced effects and so on, and as such represents a whole new study. All good studies offer new interesting questions. All great ones, even more! Nevertheless, it does tend to stand out as the missing data set, especially since it is the centrepiece of the argument.

My concern at the absence of an IH group primarily arises from the choice of model used in the study. It is certainly novel. And the results arising from the cMPT lesions are fascinating. However, the authors overplay the relevance of the model and would appear to protest too much in respect of its natural value.

It is not at all clear that the lesions cause airway obstruction. This was not demonstrated. Therefore, I suggest that the title and throughout the manuscript should refer to sleep-disordered breathing, not OSA. The title is potentially misleading. Even then, this needs to be clarified. It is evident that cMPT lesion causes sleep disruption, apparent sleep-disordered breathing, and hypoxaemia. There is evidence too of cBF neuronal loss and cognitive impairment. Although stronger arguments for the pathological mechanism can be made based on data presented later in the manuscript, at first pass it is difficult to dissociate sleep-disruption from sleep-disordered breathing and hypoxaemia. Studies using WBP could be better explored. For example, it is not clear why mice are hypoxaemic if ventilation is unaffected. Automated analysis of breathing might not adequately reveal periods of hypoventilation and apnoeas. Some mice appear to have stable hypoxaemia, whereas others appear to show intermittent hypoxaemia (reflective of dysregulated breathing in sleep). The enhanced pause metric is generally criticised as inadequate by respiratory neurobiologists using WBP. The authors should score the respiratory traces and in particular count periods of apnoeas and hypopnoeas. They should report data for wakefulness. Were lesioned mice normoxic during resting waking? A factor relevant to consideration of the model is that the lesion likely affected sleep, and as such when

comparing control v lesioned animals, breathing is perhaps being assessed during different epochs in the 2 groups. The authors simply refer to "sleep" and should reveal how like-with-like assessments were performed. In this way, it may be differences in sleep architecture/ sleep pressure that reflect differences in breathing, since breathing differs naturally in the different phases of sleep. If chronic sleep disruption in lesioned animals causes more REM sleep in lab-based assessment of breathing, then REM-related breathing episodes (dysregulated breathing) will be more frequent in lesioned animals, which does not necessarily mean that disordered breathing is a primary outcome per se. Less REM and more non-REM might be associated with hypoventilation and hypoxaemia. Inadequate time to capture all phases might contribute to considerable heterogeneity in the assessments in lesioned animals. I should like to learn more about sleep and breathing in this model.

The authors ascribe sleep disruption in lesioned animals to sleep-disordered breathing, that is, arousals because of (presumed) airway obstruction, and at least SDB, which it is posited drive sleep fragmentation. This certainly is the case in OSA, but it is not evident in this model and cannot be convincingly concluded based on the data and design. Admittedly, it is informed in part by the evidence that oxygen supplementation prevents cBF cell loss and sleep debt in lesioned animals, suggesting that hypoxaemia is a driving force. But high oxygen does much more than just reverse the potential for low oxygen.

Breathing should be scored and Poincare plots for breath to breath variability would be useful.

A concern is that recording sessions were very short and so it is difficult to conclude what the respiratory phenotype of the lesioned mice really is. Displaying the temporal relationship between breathing and SpO₂ might better reveal that there are true periods of hypoventilation/apnoea/dysregulated breathing, although again whether this is a direct result of the lesion or second to disrupted sleep (given the role of cMPT neurons) is not clear. The authors refer to fluctuating hypoxaemia, but it appears relatively stable in some mice albeit over the very short recording shown.

I also don't understand the authors' premise that cMPT lesion would cause airway obstruction. It is established that REM-related hypoglossal hyperpolarisation (a precursor to airway collapse due to muscle atonia) is caused by cholinergic transmission at the XII motor nucleus (Horner group, Toronto). So disruption of cholinergic input to XII motoneurons might be expected to disinhibit XII motor outflow. It is also not clear that apnoeas in rodents are obstructive events and without an index of respiratory effort, the authors cannot distinguish central and obstructive events. It is also not clear that there are apnoeas. For all of the limitations of WBP, apnoeic events can be clearly distinguished on the respiratory flow trace. Manual scoring of events is advised. For these reasons, in terms of the model, it seems more intuitive to consider that this is an animal model of sleep disruption with associated disrupted breathing (though this needs to be shown more convincingly) and apparent hypoxaemia (which could be persistent or intermittent). A drawback to the study is the lack of sleep-staging during WBP. Also, whereas preventing hypoxia is later shown to be protective, how is sleep 'normal' in lesioned animals given the role of cMPT neurons in REM sleep. Perhaps it isn't, but high oxygen prevents sleep-related disruptions in breathing, consequential IH and as such blocks the driving force as argued by the authors?

However, for all of the above critique, the outcomes in the model provided the authors with a testable hypothesis. Namely, that hypoxaemia arising in this model, as it does in other conditions, might be the driving force in the elaboration of cBF neuronal loss, sleep disruptions, AD pathology and worsened behaviour/cognition.

The studies that follow the introduction of the model are informative. Sleep deprivation alone does not cause loss of cBF neurons. AD pathology is not worsened. Daily sustained hypoxia (authors should clarify that 80% relates to SpO₂ measurements; What was the environmental oxygen concentration used?) was not sufficient to replicate key morbidities in the lesion model, suggesting that pattern of hypoxia is a key determinant as has been established for several other IH-dependent (and OSA relevant) morbidities. The intervention studies add considerable value to the study pointing to p75-

NTR_HIF-1a dependent neuronal cell loss which perpetuates AD pathology.

For all of the elegance of the study, I'm not convinced that cPMT lesion causes SDB which causes sleep disruption. I'm not convinced that there is a clear IH phenotype in the model. The hypoxaemia in the model is not clear, but clearly 8h per day at 80% SpO2 does not cause the phenotype. I am convinced that lesion causes a p75NTR-HIF-1a mediated loss of cBF neurons, which appear important to several cognitive behaviours. Oxygen supplementation is also remarkably beneficial, but oxygen supplementation is not just reversal of hypoxaemia (especially at 40%!). Although it seems logical to link all of this to IH, the link comes from other work and is not entirely evident in the paper. I can't help feeling that there is something else at play in this model.

I don't agree with the use of terminology such as authentic model, naturalistic model etc. All models are just that and have strengths and weaknesses. Selective lesion of cMPT is not a natural model of OSA.

I strongly urge the authors to re-consider the analysis and presentation of the WBP data. The paper would also be considerably strengthened by the addition of IH-exposed group(s).

Reviewer #4 (Remarks to the Author):

The link between obstructive sleep apnea (OSA) and Alzheimer's disease (AD) risk has been suggested by epidemiological studies for a number of years but has been hard to demonstrate in human for a number of reasons: i) the link between OSA and AD is likely bidirectional; ii) test of causality via treatment with CPAP is hard to evaluate due to the length of the clinical trials that would be needed and the general poor compliance to treatment; iii) both OSA and AD may have to some extent, a shared unknown etiology associated with aging, or a shared known morbidity like vascular-related factors. Modelling OSA in mice with the methods proposed has the advantage of providing a naturalistic model of testing the causality of these associations without the confounding effects of obesity or hypertension and has multiple advantages which adds to the highly innovative aspect of this manuscript which I recommend to publish without reservations. The study design and results are incredibly promising and important for this emerging field. All minor comments:

Introduction: The Elias paper is in fact a negative finding (the effects were driven by ApoE4 carriers), please cite instead the recently published paper by André C et al on JAMA Neurology or the Sharma R Blue journal paper instead. Please add a sentence or two of the current limitation of animal models of OSA and the gaps that the current model fills (the comparison to chronic hypoxia and lack of findings adds to the merits of this novel cMPT model).

Methods: please describe why the light cycle was chosen to perform all behavioral experiments, and discuss whether this could be a limitation.

Results: do cMPT lesion animals spend the same percentage of time on REM sleep?

Discussion: although the cMPT model is one of the most relevant contributions to the field I have seen over the past years, please add as a limitation that this model only replicates some of the key features of OSA in humans and perhaps cite some of the exciting work from the Eckert lab on the different OSA phenotypes.

Whenever possible I'd recommend to avoid referencing to review papers and cite the original contributions to the findings.

REVIEWER COMMENTS

Reviewer #1 (Remarks to the Author):

This interesting study aimed to explore the impact of obstructive sleep apnea (OSA) combined with death of cholinergic neurons in basal brain nuclei on the progress of Alzheimer's disease (AD) in engineered model mice carrying a mutated human beta amyloid gene and subjected to Saporin injection for destroying the p75 receptor and exacerbating the deterioration of deep cholinergic nuclei. The topic is of great interest to many, and previous reports were less authoritative than hoped since the employed model systems were not optimal. In the current manuscript, the authors describe a novel OSA model based on much effort and care that was established as precisely as possible, which is a major strength point of this study. While the induced disease in these mice involves both a human beta amyloid transgene and impaired sleep, and although the injected saporin affects the murine brain, avoiding the OSA diminishes the exacerbated disease symptoms in tested mice. This provides experimental support to the authors' working hypothesis and strengthens their claims for the relevance of using air pressure therapy for delaying the progress of familial AD. However, this study also entails several key points which raise key questions and call for additional experimental work, as is detailed below.

Major comments

1. While the abstract describes 'oxygen-sensitive p75 receptors' as causally involved in the OSA-induced exacerbation of AD, the basal nuclei p75 receptors in the mesopontine nuclei of the currently studied mice were destroyed by saporin injection; although suppressing the OSA showed its causal relevance in the observed symptoms, the saporin poisoning raises a question regarding the role of p75 receptors in basal brain nuclei in the impact of OSA and must be declared in the abstract and referred to in the results and Discussion sections in larger detail.

There is no role for p75 receptors in the mesopontine neurons in our sleep disordered breathing (SDB) mouse model.

- (i) cholinergic mesopontine neurons in the brain stem do not express p75.
- (ii) The saporin compound used in the study was a novel urotensin-2 peptide-conjugated saporin (UT2-SAP). Urotensin-2 receptors are expressed uniquely by the cholinergic mesopontine neurons. The reviewer may have assumed we used the anti-p75 antibody conjugated-saporin commonly used to kill cholinergic basal forebrain (cBF) neurons that express p75, which does not kill the cholinergic mesopontine neurons.
- (iii) We are also comfortable that there is no direct effect of UT2-SAP on cBF neurons. Two weeks after the injection, mesopontine neurons die, but the number of p75-positive cBF neurons is normal, even though they subsequently die. We demonstrate the cBF but not cMPT neurons are protected after various treatments (high oxygen, HIF1 α inhibitor and – results not included here - knockout of p75), which would not be the case if the saporin was targeting them.

To address this issue, we have focussed on HIF1 α - mediated mechanisms of cell death rather than evoke p75-induced pathways to ensure that there is reduced possibility for any confusion.

2. The study involved both male and female mice and does not distinguish between sexes. However, ample reports demonstrate that women are more readily susceptible to develop AD, and that they deteriorate faster when the disease progresses even when their longer life expectancy is factored in. However, the underlying mechanisms are still unknown, and the current study raises the question of OSA might be causally involved. Therefore, performing the reported tests separately in male and female mice, and comparing their susceptibility to OSA based on experimental evidence would be most valuable.

As we are both inducing sleep disordered breathing (SDB) by direct lesion and doing so in genetically programmed mice, this study cannot assist in understanding gender differences related to SDB the aetiology in AD. Nonetheless, we have clarified our methods and results where data are from mixed gender cohorts or from a single gender. There are differences in human breathing patterns, e.g. reduced airway diameter and lung volume in females which result in lower peak expiratory flow and vital capacity. Similarly we find that the breathing patterns during exhalation of male (**new Fig 2E**) and female (**new Sup Fig 2**) mice differ independently of sleep, particularly in average measures of total time to exhale (EEP; $p=0.0212$ sleep-wake-paired, two way ANOVA) and for the point in expiration where the peak flow occurs as a fraction of total time to expire (Rpef, $p=0.051$); the averages of these measures in control male mice were significantly greater during sleep than those of females, and female mice exhibited more breaths per minute. Although our results are similar between wildtype male and female mice (with main effects being lesion group or sleep state), we predominantly analysed male mice in the subsequent studies when using wildtype cohorts as their sleep-disordered breathing phenotype was more pronounced. However, due to the availability of aged animals, both male and female AD model mice were used; no differences between genders were observed with regards to the size or number of plaques or in behaviour using a 2 way ANOVA (with the main effect being lesion).

3. Considering the topic of this study and the recently proved importance of specific muscarinic receptors for REM sleep, quantifying the levels of such receptors and/or their coding transcripts and citing the corresponding Ueda references may offer a molecular link and a more precise measure of the affected processes.

We feel that whether or not particular muscarinic receptors are needed to control sleep/wake cycles is tangential to our findings, albeit interesting. In our model, the sleep disruption is due to the cMPT lesion and resulting hypoxemia as this phenotype is resolved by high oxygen treatment. It is possible that the subsequent loss of cBF neurons affects cholinergic neurotransmission, as a result of whether muscarinic receptors may be up- or down-regulated, but it is less likely to be a specific receptor subset. However, we have added new data which demonstrate that the extent of ChAT immunofluorescent staining in cBF axons in the cortex is reduced, indicating cholinergic neurotransmission is indeed likely to be decreased (**new Sup Fig 5C**).

4. Apart from the deep nuclei, AD involves escalated thinning of the cortical layer where much of the cholinergic input is provided by VIP-ACh interneurons. Quantifying these neurons as well could add another causal element to the study of OSA-induced AD and should be tested.

The ChAT-positive interneurons in the cortex are not known to express the urotensin 2 receptor nor are they sufficiently closely located to the brain stem lesion site to be expected to be killed by our lesion method. It is possible that subsequent loss of these (or other) neurons does occur due to the MPT loss (as we report for the cBF neurons), but we are not aware of a reason why cortical VIP neurons would be specifically vulnerable compared to other cortical interneurons or pyramidal neurons. Nonetheless, to address this issue we first quantified the width of the cortical and hippocampal layers in control and cMPT-lesioned AD model mice as an indication of widespread degeneration. No differences were found (**new Fig 5JK**), indicating that at the timepoint analysed there was no substantial cortical degeneration (as observed in human AD aetiology; Schmidt et al., *Cell Reports* 2016). As suggested, we then undertook a pilot study to assess the number of ChAT-positive cortical interneurons in cMPT-lesioned and control mice. Given the small number of neurons per section (in mice, the cholinergic VIP neurons are a very small proportion (~15% of the total VIP neuron proportion and <1% of all cortical neurons) we did not observe a difference between conditions. A larger mouse cohort would be required to be appropriately powered to detect any loss smaller than a 50% difference, which we do not believe is justified in this study.

5. The navigation learning capacities of tested mice appear to be only partially damaged, possibly indicating that OSA affects hippocampal functioning less dramatically than its impact on other brain regions. In this context, measuring changes in hippocampal volume in the affected mice may be of interest.

We have now provided evidence that cholinergic neurotransmission is likely reduced in the cortex and hippocampus (**New Sup Fig 6C**). However, as described above (point 3), we also report that there is no change in the width of the cortical and hippocampal layers in control and cMPT-lesioned AD model mice (**new Fig 5J,K**), indicating that at the timepoint analysed there was no substantial degeneration of these brain areas. As cholinergic innervation is partially disrupted, it is not surprising that the spatial memory abilities of the mice remain partially intact.

6. Given the importance of neuroinflammation for the progress of AD, quantifying the neuroinflammation extent should be presented in the main body of this manuscript and involve in-depth statistics, again separately in male and female animals.

We have undertaken additional statistical testing comparing male and female mice for inflammation markers but did not find a sex effect (**new Sup Fig 4**).

7. To more substantially prove the causal involvement of HIF1 α in the damaged CMPT neurons, one should test the impact of these neurons on the OSA-induced exacerbation of AD progression with and without interruption of HIF1 α expression in CMPT neurons.

We apologise of any confusion; we do not propose that HIF1 α regulates the damage to cMPT neurons (which are killed by UT2-saporin which is a ribosome inhibitor). Rather we propose that HIF1 α expression is required for the subsequent loss of cBF neurons. To address this directly, we undertook experiments using a conditional heterozygous HIF1 α knockout mouse in which cholinergic neurons have only 1 copy of the HIF1 α gene. The extent of cMPT neuronal loss due to the lesioning was significantly reduced compared to that in control lesioned mice, and similar between wild type and HIF1 α gene knockout mice. However the number of cBF neurons in the HIF1 α knockout mice was equivalent to that in unlesioned littermate mice. These new data are now presented in **Fig 7 I**. Together with our data assessing HIF1 α nuclear localisation in cBF neurons, our results indicate that an increased expression/accumulation of HIF1 α within the cBF neurons is required for their degeneration. As cBF lesioning exacerbates AD pathology in these mice, we anticipate that the rescue of cBF degeneration by HIF1 α inhibition would rescue the phenotype, similar to the high oxygen treatment rescue. Due to the time and cost it would take to generate a triple transgenic aged AD mouse lacking HIF1 α we were unable to determine the effect of HIF1 α depletion on AD pathology.

8. Considering the recent publications reporting an increased risk of dementia in aged patients under chronic anti-cholinergic medications, it is logical to quantify the cholinergic tone in these mice, possibly by measuring cholinesterase activities in the circulation.

The reviewer has raised another interesting point regarding the aetiology of cholinergic neuron degeneration as a risk factor for dementia. Our main message is that environmental factors such as OSA/SDB, may exacerbate AD development because of their effects on cBF neurons (and in the case of the latter, effects on cMPT neurons as well). Anti-cholinergic medications may also affect AD development due to cholinergic loss – as occurs when one lesions cBF neurons in AD mice (e.g. Hampel 2002). Rather than specifically assessing cholinergic tone, we added quantification of cholinergic axonal ChAT expression, which indicates a loss of neurotransmission is probable in our model, which could be extrapolated to mimic anti-cholinergic medications (in the absence of SDB).

9. Apart from OSA patients, this study may infer added value for treatment with high oxygen levels as delaying AD symptoms severity. This needs to be discussed.

In our experiments we used high oxygen treatment as a means to prevent hypoxia/hypoxemia, with

the treated SDB mice having 95% or higher blood oxygen levels (SpO₂). The direct equivalent treatment for humans would be sleeping in a hyperbaric chamber, but high levels of oxygen is not necessarily healthy (e.g. it can result in oxidative stress). We have added this point to the discussion.

Minor comments

1. The cited Hardy and Selkoe review in the introduction is almost 20 years old; wasn't there anything fresher which is worth citing since 2002?!

The cited review by experts in the field outlines the classic amyloid hypothesis and the relationship between the genetics and disease, which, although old, was appropriate for the point where it was cited. Nonetheless we have added a recent review outlining the current state of the hypothesis. Eric Karran & Bart De Strooper (2022) The amyloid hypothesis in Alzheimer disease: new insights from new therapeutics, *Nat Rev Drug Discov.* doi: 10.1038/s41573-022-00391-w.

2. The experimental use of whole body plethymography has been pivotal for reaching the current conclusions, and may be mentioned in the title.

Although whole body plethymography is a powerful method we feel it is but one of several methods we used and therefore chose not to include it in the title due to space considerations. We are happy to take advice from the editor regarding this or any other title changes.

3. To ensure maximal impact of the observed phenomena, the data availability should be open, even if in a local website; and not dependent on personal requests from the corresponding author.

Our data will be available on a server with a permanent DOI, as is preferred by our Governmental funding guidelines. Once allocated no data or files can be changed without a new DOI being created so will wait for our manuscript to be 'in press' before publishing the data.

4. Employing commercially available mice with fluorescently labeled ChAT expressing neurons may offer a better follow-up of the loss of such cells in the mesopontine nuclei.

We have taken this idea as a suggestion for our ongoing and future work.

5. Adding more information on the OSA parameters in human patients may provide a valuable comparison to the current study and increase its biomedical value.

More information on the OSA/SDB parameters in human patients has been added to the Results where we have added new sleep and breathing data (**new Figs 2 and 3**) and in the Discussion

7. The added value compared to the Shiota et al., 2013 article should be highlighted.

The value of our findings beyond replicating those of Shiota et al with a different model, are: (i) we have shown that the SDB model includes sleep disruption, and breathing is more variable/naturalistic of SDB compared to the highly regulated oxygen level fluctuations applied to mice in intermittent hypoxia chambers; (ii) the cBF neuronal death and exacerbation of plaque, inflammation, cognitive impairment pathology was rescued by preventing hypoxia in our OSA model, and (iii) the mechanism of cBF neuronal death is dependent on the accumulation and nuclear translocation of HIF1 α within these vulnerable neurons. These points have now been emphasised in the Introduction and in the Discussion.

8. Fig 2H presents data of individual mice which should be replaced by average group values in male and female animals.

We have now added new data regarding breathing measures (**new Fig 2**). These were not able to be combined due to differences in baseline breathing frequencies between sexes (**new Sup Fig 2**).

9. Fig 1, 4, 7 should all include size bars in all pictures, with enlarged cells shown for better identifying the presented differences. Correction of three dimensional errors is recommended.

We have ensured that all photographic images now have size bars and have updated images where sizes were small. Our cell counts are not performed by stereology but all ChAT-positive cells in a section in the region of interest defined by the Mouse Brain Atlas, and all sections containing the basal forebrain or mesopontine structure (within a 1 in 3 series) are counted per animal, with the raw cell numbers per mouse graphed. 3D errors are therefore not required to be corrected. This has now been further explained in the Methods.

10. Supplementary Fig 5 shows reduced astrogliosis in mice under OSA, which is confusing as it might reflect lower stress levels. This should be explained.

The reduced astrogliosis (**Sup Fig 7G-J**) in OSA/SDB AD mice that were given high O₂ treatment was assumed to be due to the coincident reduced A β accumulation in the brain sections (**Fig 7C,D**). However, it is also possible that if the mice were no longer struggling with breathing, they might have had reduced stress that contributed to this phenotype, a possibility that has been added to the Results.

Reviewer #2 (Remarks to the Author):

Qian and colleagues conduct a very interesting study using a mouse model to establish a link between sleep apnea and Alzheimer's pathology. Two factors mediate the pathological changes: oxygen sensitive p75 neurotrophin receptor and hypoxia inducible factor 1 alpha. Sleep apnea was induced by producing cMPT neuron lesions, using UII-SAP injected directly into the LDT and intraventricularly. Oxygen therapy prevented the formation of the Alzheimer's pathology. Overall, the studies seem well conducted and provide good evidence for a physiological mechanism through which sleep apnea can lead to Alzheimer's disease.

My primary concern about the study is the statistical analyses. The statistical analysis section offers no discussion of how the data are analyzed. There is no mention of statistical tests or methods used to compute confidence intervals. No confidence intervals are reported. The statistical analyses entail primarily comparisons of the differences in two independent samples, suggesting two-sample, t-tests, z-tests and/or Mann-Whitney non-parametric test. Most of the samples are small. The largest samples have 15 animals per group. Reporting p-values is a very weak approach for making inferences because all they say, when results are found to be significant, is that the null hypothesis improbable. Two things should be done here. The authors should make their comparisons by constructing confidence intervals for the median differences between the groups. The inference is made, i.e. the two groups would be considered different, if the 95% or 99% confidence intervals do not contain zero. The important added advantage of the confidence intervals is that they make it possible to measure directly the magnitude of the differences between the two groups and the uncertainty in those differences. To really explore the variability in the data these confidence intervals should be computed by non-parametric bootstrap methods. If analyses are redone in this manner, I suspect that several of the results which are reported here as significant by the current analyses will no longer be significant. The authors should then reassess the validity of their inferences. It may be necessary to change the conclusions and or conduct more studies. The statistical tests are mentioned in the figure legends. They should be discussed in the Methods section of the paper.

We have recalculated all our statistics and have expanded the statistical section of the Methods. We have also provided the confidence intervals for the majority of the figures in a Reviewers' Appendix of figures available until 18 July 2022 here:

<https://cloudstor.aarnet.edu.au/sender/?s=download&token=8e23b14a-b89e-4ad7-80c9-8febe6552e6e>. None of our conclusions changed based on the use of confidence interval statistics.

As we are less used to reporting and interpreting confidence intervals, the majority of our main figures remain presented as mean with standard error, and individual measured indicated. We would appreciate advice from the reviewers and editor as to which presentation is most appropriate should the manuscript be accepted.

Reviewer #3 (Remarks to the Author):

This is a very interesting, often fascinating, study in a topical and important area of investigation. Several approaches taken employ state-of-the-art experimental techniques. Several data sets are very convincing and undoubtedly support the conclusions drawn. However, there would also appear to be some short-comings and conclusions that are over-stated and/or confusing (at least to me).

I commend the authors on a series of experiments that are elegant and carefully conducted. The data are reported in transparent fashion. There is a coherent basis to the sequence of experiments performed, culminating in some fascinating observations that provide mechanistic insight to p75NTR-HIF-1 α -dependent neuronal cell loss implicated in exacerbation of AD pathology.

I was less convinced by some of the data and study design in early experiments, that make it difficult to determine if disrupted sleep causes disrupted breathing or vice versa in this new model. In my view, there are some problems here that ought to be acknowledged in the manuscript. There are other experimental approaches that would add considerable rigor.

However, the studies illustrating convincingly that 2ME-2 treatment, or p75NTR selective knockout, or oxygen therapy each prevent cBF neuronal cell loss and impaired cBF-dependent behaviours (or at least some of them) are impressive.

A slight worry is that the authors did "everything" but administer intermittent hypoxia, which it is argued, is the driving pathological feature (and undoubtedly is relevant to OSA). Perhaps it is because this has already been done (Row et al.2007)showing CIH-induced cBF neuronal cell loss and cognitive impairment. This old paper rather steals from the major conclusion of the present study in that the argument has been made already. The present study extends the argument to reveal a potentially causal signalling pathway and possible druggable target. Since the authors are well placed to perform studies of intermittent hypoxia (given the study of daily sustained hypoxia), the final piece in the puzzle i.e. showing that IH drives pathology via p75NTR-HIF-1 α is noticeably absent. I highlight this because although the argument can be readily made it is not entirely conclusive from the collective data in the current study.

Of course, I recognise that this in turn might provoke interest in the efficacy of 2ME-2 or disruption to p75NTR in ameliorating IH-induced effects and so on, and as such represents a whole new study. All good studies offer new interesting questions. All great ones, even more! Nevertheless, it does tend to stand out as the missing data set, especially since it is the centrepiece of the argument. My concern at the absence of an IH group primarily arises from the choice of model used in the study. It is certainly novel. And the results arising from the cMPT lesions are fascinating. However, the authors overplay the relevance of the model and would appear to protest too much in respect of its natural value.

It is not at all clear that the lesions cause airway obstruction. This was not demonstrated.

Therefore, I suggest that the title and throughout the manuscript should refer to sleep-disordered breathing, not OSA. The title is potentially misleading. Even then, this needs to be clarified.

It is evident that cMPT lesion causes sleep disruption, apparent sleep-disordered breathing, and hypoxaemia. There is evidence too of cBF neuronal loss and cognitive impairment. Although stronger arguments for the pathological mechanism can be made based on data presented later in the manuscript, at first pass it is difficult to dissociate sleep-disruption from sleep-disordered breathing and hypoxaemia. Studies using WBP could be better explored. For example, it is not clear why mice are hypoxaemic if ventilation is unaffected. Automated analysis of breathing might not adequately reveal periods of hypoventilation and apnoeas. Some mice appear to have stable hypoxaemia, whereas others appear to show intermittent hypoxaemia (reflective of dysregulated breathing in sleep). The enhanced pause metric is generally criticised as inadequate by respiratory neurobiologists using WBP. The authors should score the respiratory traces and in particular count periods of apnoeas and hypopnoeas. They should report data for wakefulness. Were lesioned mice

normoxic during resting waking? A factor relevant to consideration of the model is that the lesion likely affected sleep, and as such when comparing control v lesioned animals, breathing is perhaps being assessed during different epochs in the 2 groups. The authors simply refer to "sleep" and should reveal how like-with-like assessments were performed. In this way, it may be differences in sleep architecture/ sleep pressure that reflect differences in breathing, since breathing differs naturally in the different phases of sleep.

If chronic sleep disruption in lesioned animals causes more REM sleep in lab-based assessment of breathing, then REM-related breathing episodes (dysregulated breathing) will be more frequent in lesioned animals, which does not necessarily mean that disordered breathing is a primary outcome per se. Less REM and more non-REM might be associated with hypoventilation and hypoxaemia. Inadequate time to capture all phases might contribute to considerable heterogeneity in the assessments in lesioned animals. I should like to learn more about sleep and breathing in this model.

The authors ascribe sleep disruption in lesioned animals to sleep-disordered breathing, that is, arousals because of (presumed) airway obstruction, and at least SDB, which it is posited drive sleep fragmentation. This certainly is the case in OSA, but it is not evident in this model and cannot be convincingly concluded based on the data and design. Admittedly, it is informed in part by the evidence that oxygen supplementation prevents cBF cell loss and sleep debt in lesioned animals, suggesting that hypoxaemia is a driving force. But high oxygen does much more than just reverse the potential for low oxygen. Breathing should be scored and Poincare plots for breath to breath variability would be useful.

A concern is that recording sessions were very short and so it is difficult to conclude what the respiratory phenotype of the lesioned mice really is. Displaying the temporal relationship between breathing and SpO₂ might better reveal that there are true periods of hypoventilation/apnoea/dysregulated breathing, although again whether this is a direct result of the lesion or second to disrupted sleep (given the role of cMPT neurons) is not clear. The authors refer to fluctuating hypoxaemia, but it appears relatively stable in some mice albeit over the very short recording shown.

I also don't understand the authors' premise that cMPT lesion would cause airway obstruction. It is established that REM-related hypoglossal hyperpolarisation (a precursor to airway collapse due to muscle atonia) is caused by cholinergic transmission at the XII motor nucleus (Horner group, Toronto). So disruption of cholinergic input to XII motorneurons might be expected to disinhibit XII motor outflow. It is also not clear that apnoeas in rodents are obstructive events and without an index of respiratory effort, the authors cannot distinguish central and obstructive events. It is also not clear that there are apnoeas. For all of the limitations of WBP, apnoeic events can be clearly distinguished on the respiratory flow trace. Manual scoring of events is advised. For these reasons, in terms of the model, it seems more intuitive to consider that this is an animal model of sleep disruption with associated disrupted breathing (though this needs to be shown more convincingly) and

apparent hypoxaemia (which could be persistent or intermittent). A drawback to the study is the lack of sleep-staging during WBP. Also, whereas preventing hypoxia is later shown to be protective, how is sleep 'normal' in lesioned animals given the role of cMPT neurons in REM sleep. Perhaps it isn't, but high oxygen prevents sleep-related disruptions in breathing, consequential IH and as such blocks the driving force as argued by the authors?

However, for all of the above critique, the outcomes in the model provided the authors with a testable hypothesis. Namely, that hypoxaemia arising in this model, as it does in other conditions, might be the driving force in the elaboration of cBF neuronal loss, sleep disruptions, AD pathology and worsened behaviour/cognition.

The studies that follow the introduction of the model are informative. Sleep deprivation alone does not cause loss of cBF neurons. AD pathology is not worsened. Daily sustained hypoxia (authors should clarify that 80% relates to SpO2 measurements; What was the environmental oxygen concentration used?) was not sufficient to replicate key morbidities in the lesion model, suggesting that pattern of hypoxia is a key determinant as has been established for several other IH-dependent (and OSA relevant) morbidities. The intervention studies add considerable value to the study pointing to p75-NTR_HIF-1a dependent neuronal cell loss which perpetuates AD pathology.

For all of the elegance of the study, I'm not convinced that cPMT lesion causes SDB which causes sleep disruption. I'm not convinced that there is a clear IH phenotype in the model. The hypoxaemia in the model is not clear, but clearly 8h per day at 80% SpO2 does not cause the phenotype. I am convinced that lesion causes a p75NTR-HIF-1a mediated loss of cBF neurons, which appear important to several cognitive behaviours. Oxygen supplementation is also remarkably beneficial, but oxygen supplementation is not just reversal of hypoxaemia (especially at 40%!). Although it seems logical to link all of this to IH, the link comes from other work and is not entirely evident in the paper. I can't help feeling that there is something else at play in this model.

I don't agree with the use of terminology such as authentic model, naturalistic model etc. All models are just that and have strengths and weaknesses. Selective lesion of cMPT is not a natural model of OSA.

I strongly urge the authors to re-consider the analysis and presentation of the WBP data. The paper would also be considerably strengthened by the addition of IH-exposed group(s).

The reviewer has provided a detailed commentary during which they raised major uncertainty or criticisms, and some more minor matters for correction or consideration. These are distilled into 3 major point and responded to below.

1. Their first major point is whether disrupted sleep causes disrupted breathing or vice versa in our model, whether there is a clear IH phenotype, and whether the lesions actually cause airway obstruction

We have provided new data (**Fig 2**) using whole body plethysmography with coincident EEG that demonstrate that the cMPT-lesioned mice have altered breathing during sleep, whereas breaths were equivalent between lesioned and unlesioned mice during wakefulness.

Unfortunately our plethysmography apparatus does not provide a breathing trace, only metrics in 2 second epochs. Therefore the requested Poincare plots are not an appropriate analysis for our data. However, the metrics pertaining to an average breath in the lesioned group indicate smaller and shallower breaths, but with more significant variability in the inhalation metrics and the time between breaths in lesioned compared with control mice i.e. periods of low breathing rates followed by periods of quite high frequency breathing. There was no significant difference between control and lesioned groups in the number of times during sleep that a 2 second recording epoch had a frequency below 100 breaths/min (indicative of missed breaths). Nonetheless, cMPT-lesioned mice were significantly more likely to have adjoining epochs of reduced frequency ($p < 0.0096$ one-way t-test) with 54% and 17% of low frequency epochs lasting longer than 4 seconds in cMPT-lesioned and control mice, respectively. Furthermore, there was a trend for the long low-frequency breathing periods to be followed by an awake epoch in the lesioned mice, whereas there was no obvious association with low frequency breathing epochs and sleep/wake transitions in the control mice. These data are consistent with the hypothesis that the mice have SDB and either hypopnea or apnoea events, but are not necessarily indicative of a causal relationship.

The new EEG and movement data (**Fig 3 new and previous panels**) further indicate that sleep was significantly disrupted, resulting in significantly more transitions between REM and non-REM sleep, in addition to the increase in awake-time sleepiness. We cannot rule out that cBF degeneration contributes to the sleep changes, but feel that this is countered by our previous data (Hamlin et al., 2014) that cBF lesions do not result in alterations in Y maze performance unlike

cMPT lesions and sleep disruption. Furthermore, we have previously reported that cBF-lesioned mice tend to take longer to rest in the light cycle, and are more active in the dark/active cycle. These points have been added to the Discussion.

2. Request for direct evidence that IH causes cBF degeneration

As acknowledged by the reviewer the requested experiment has been published previously by others. Therefore, as we have supplied additional data supporting a disrupted breathing phenotype during sleep (**new Figs 2 and 3**), we argue that these data are not essential for our conclusions and publication.

Nonetheless, we acknowledged that it would ‘round off’ our manuscript to have a condition of intermittent hypoxia and attempted to provide it. The application of intermittent hypoxia to caged mice requires a specialised gas control system and small cages to enable fast exchange of gases e.g. every 90 seconds. This equipment is not available at our University. We therefore initiated a collaboration with a group in another state (who uses the equipment to study heart ischemia). We were successful in undertaking a pilot study using the tissue from the controls of the transgenic strain of mice that they were studying; this was to test the practicalities of shipping the brain sections to us for analysis. However, the number of cBF neurons in their control animals (and intermittent hypoxia animals) was significantly lower than in our wildtype mice. Subsequently due to COVID-19 shutdowns in their city and related disruptions to their research programs, and the need to troubleshoot the tissue dissection and processing, they were unable to prioritise our experiments. When resources allow, we aim to undertake such studies in which the depth and length of hypoxemia and reperfusion required to induce cBF degeneration are investigated using a controllable system as a comprehensive (but follow-up) study.

3. Wording regarding our model as naturalistic..., as a result of obstructive sleep apnea, or due to intermittent hypoxia.

In response to this reviewer’s points, and our additional results, we have revised our framing of the model and altered the use of OSA to SDB throughout the manuscript. We have also clarified that the % oxygen saturation is SpO₂ within the text.

Reviewer #4 (Remarks to the Author):

The link between obstructive sleep apnea (OSA) and Alzheimer's disease (AD) risk has been suggested by epidemiological studies for a number of years but has been hard to demonstrate in human for a number of reasons: i) the link between OSA and AD is likely bidirectional; ii) test of causality via treatment with CPAP is hard to evaluate due to the length of the clinical trials that would be needed and the general poor compliance to treatment; iii) both OSA and AD may have to some extent, a shared unknown etiology associated with aging, or a shared known morbidity like vascular-related factors. Modelling OSA in mice with the methods proposed has the advantage of providing a naturalistic model of testing the causality of these associations without the confounding effects of obesity or hypertension and has multiple advantages which adds to the highly innovative aspect of this manuscript which I recommend to publish without reservations. The study design and results are incredibly promising and important for this emerging field. All minor comments:

Introduction: The Elias paper is in fact a negative finding (the effects were driven by ApoE4 carriers), please cite instead the recently published paper by André C et al on JAMA Neurology or the Sharma R Blue journal paper instead.

Thank you for finding this erroneous reference and suggesting appropriate replacements, which have been made.

Please add a sentence or two of the current limitation of animal models of OSA and the gaps that the current model fills (the comparison to chronic hypoxia and lack of findings adds to the merits of this novel cMPT model).

We have amended the end of the introduction to read “Current animal models of SDB are limited to modelling chronic hypoxia, artificially induced intermittent hypoxia through rapid cycling of air in a small chamber with varying oxygen concentrations, or sleep disruption. In order to understand the mechanisms linking AD with OSA we developed a method to model SDB in mice in which breathing and sleep were disrupted coincidentally but in the absence of comorbidities.”.

Methods: please describe why the light cycle was chosen to perform all behavioral experiments, and discuss whether this could be a limitation.

We have revised the methods to read: Behavioural experiments were always performed in the light cycle. Although this was during the animals' sleep cycle, we reasoned, that this timing permitted sleep-deprived animals to have caught up any lost sleep prior to the testing. (Methods, pg 5)

Results: do cMPT lesion animals spend the same percentage of time on REM sleep?

Surprisingly, EEG recordings have revealed that cMPT-lesioned mice have increased REM sleep and decreased non-REM sleep, due to increased transitions between these two sleep states, but that overall they have the same sleep time as sham-lesioned mice. These findings are presented in the new Figure 3.

Discussion: although the cMPT model is one of the most relevant contributions to the field I have seen over the past years, please add as a limitation that this model only replicates some of the key features of OSA in humans and perhaps cite some of the exciting work from the Eckert lab on the different OSA phenotypes.

We have now highlighted that “Our model will require further examination to determine whether the disordered breathing during sleep in the cMPT-lesioned mice accurately mimics a form of SDB in humans” and cited

A. Edwards, DJ Eckert, AS Jordan (2017) Obstructive sleep apnoea pathogenesis from mild to severe: Is it all the same? *Respiratory Sleep Disorders* 22:33–42

and

M Bosi, A De Vito, D Eckert, J Steier, B Kotecha, C Vicini, V Poletti (2020). Qualitative phenotyping of obstructive sleep apnea and its clinical usefulness for the sleep specialist. *Int. J. Environ. Res. Public Health*, 17: 2058

Whenever possible I'd recommend to avoid referencing to review papers and cite the original contributions to the findings.

We have tried to do this and have made some changes, but have been somewhat constricted by the number of references allowed by the journal.

REVIEWER COMMENTS

Reviewer #1 (Remarks to the Author):

Thanks for the in-depth response to my comments and the added information, which fully answer my queries.

Reviewer #2 (Remarks to the Author):

The authors have addressed my concerns regarding the statistical analysis. As I said before, the link between sleep apnea and AD seems plausible.

Reviewer #3 (Remarks to the Author):

Thank you for addressing my comments and for making revisions to the text. I commend the authors on the provision of new experiments that contemporaneously assess breathing and EEG in behaving animals allowing comparison of respiratory flow parameters during wakefulness and sleep. This provides important information on the selective expression of respiratory disturbances during sleep in lesioned animals. In particular, the demonstration of increased variability in flow-related parameters in lesioned animals only during sleep is supportive of the authors conclusion that the model elaborates a form of sleep-disordered breathing with associated hypoxaemia.

I am familiar with at least one version of the Finepoint software, which informed my previous suggestion to conduct manual analyses of the respiratory flow trace. It appears the authors have a different software package that does not permit this analysis and based on the non-standard nomenclature (provided by the manufacturer) for flow analysis it strikes me that the analysis package is likely more commonly used for flow analysis in the context of bronchoconstriction. The automated analysis likely provides information exclusively on the breaths captured for analysis but there remains the potential issue that patterns potentially excluded from analysis (by the software algorithm) do not contribute to the overall analysis of breathing pattern/instability. In short, it may be that the authors are underestimating instability in the respiratory phenotype. Details of any exclusion criteria applied by the authors in the breath-to-breath analysis (subsequently compiled in 2-s bins by the software) should be stated in the Methods. (In the version of the software I am familiar with, inclusion/exclusion criteria can be manually applied in a user-defined fashion to exclude periods of motion-induced artifact and periods of exploration/sniffing etc). Notably, the authors do not provide information on tidal volume and hence minute ventilation, and no information is provided on metabolism (which are standard in the field of respiratory control). It is understood that not everything of interest (or potential interest) can reasonably be measured in any one study. My suggestion is to include a statement in the Discussion that a thorough assessment of respiratory control assessing other aspects of the respiratory phenotype would be interesting and likely informative. For example, it would be most interesting to determine if there is increased incidence of apnoea and indeed whether this relates to central and/or obstructive events.

Interestingly, SpO₂ analysis suggests, as pointed out previously, that the lesioned mice express persistent hypoxaemia during recording sessions and not an intermittent hypoxaemia that might be expected with periods of intermittent apnoea (distinct from flow-limitation). It thus appears that the model principally elaborates a form of flow-limited sleep-disordered breathing, which in essence has been well characterised by the authors in the study.

The revised Figure 2 is clear but the corresponding text Results section (Page 7) contains errors and some inexact language that can be easily remedied.

Please refer to sex (not gender) throughout the manuscript.

Page 7: I suggest referring to flow-limitation (not resistance), or clarify that you are referring to upper airway resistance and not bronchoconstriction.

peak inspiratory "volume" (should be flow).

"Surprisingly, breathing frequencywas shorter". Should be lower. The duration of one or more periods in the cycle was shorter.

"Breath rate of 4s". Inexact. Rate is a value per unit time. So express as events per 4 s, or in other standard ways (i.e. per min).

The example of 3 breaths per 2s is incorrect. Should be 5 for 150 breaths per min.

A p value of $p < 0.0096$ is stated. Is the exact p value known? Should this be $p = 0.0096$? The text states one-way t - test. Please clarify. One way ANOVA or t test? If t-test, state whether one or two-tailed throughout the manuscript. Confirm equal variance in data sets, or otherwise apply Welch test or equivalent.

Please carefully review this whole text section. It requires revision for exact expression. I appreciate that the nomenclature may be less familiar to the authors (indeed they are non-standard in the field of respiratory control), but better clarity of expression is required so that the findings are clearly stated matching the clear presentation of the findings in Fig. 2.

It is evident that transitions between sleep states are increased in lesioned animals, and this is nicely illustrated in the revised Fig. 3.

The authors explain why it was not logistically easy to perform IH exposures and that collaborative studies were affected by COVID-related disruptions, which I fully accept.

Congratulations on the completion of a very interesting study.

Ken O'Halloran
University College Cork, Ireland.

Reviewer #4 (Remarks to the Author):

My previous critiques and comments have been addressed adequately.

Response to Reviewers

Reviewers 1, 2 and 4 had nothing for us to address

Reviewer 3

Thank you for addressing my comments and for making revisions to the text. I commend the authors on the provision of new experiments that contemporaneously assess breathing and EEG in behaving animals allowing comparison of respiratory flow parameters during wakefulness and sleep. This provides important information on the selective expression of respiratory disturbances during sleep in lesioned animals. In particular, the demonstration of increased variability in flow-related parameters in lesioned animals only during sleep is supportive of the authors conclusion that the model elaborates a form of sleep-disordered breathing with associated hypoxaemia.

I am familiar with at least one version of the Finepoint software, which informed my previous suggestion to conduct manual analyses of the respiratory flow trace. It appears the authors have a different software package that does not permit this analysis and based on the non-standard nomenclature (provided by the manufacturer) for flow analysis it strikes me that the analysis package is likely more commonly used for flow analysis in the context of bronchoconstriction. The automated analysis likely provides information exclusively on the breaths captured for analysis but there remains the potential issue that patterns potentially excluded from analysis (by the software algorithm) do not contribute to the overall analysis of breathing pattern/instability. In short, it may be that the authors are underestimating instability in the respiratory phenotype.

Our software package is quite old and we agree that it is possible that it could be excluding or including certain breathing patterns in the automatic analysis. Future use of different software with which breathing traces can be manually assessed and scored (as occurs in human analyses) would provide a more thorough characterization of the changes in breathing induced by our experimental lesions. This point has been added to our Discussion (pg 16).

Details of any exclusion criteria applied by the authors in the breath-to-breath analysis (subsequently compiled in 2-s bins by the software) should be stated in the Methods. (In the version of the software I am familiar with, inclusion/exclusion criteria can be manually applied in a user-defined fashion to exclude periods of motion-induced artifact and periods of exploration/sniffing etc).

Regarding exclusion criteria post data collection for the breath analysis, only data points for which the sleep/wake state was clear were included in the analysis, with no other subsequent exclusion criteria being applied. This is now stated in the Methods (pg 25). In one recording session we found that the data from the 12th second of each minute was replicated 10 times in the exported spread sheet; the replicated data were manually removed prior to analysis.

Notably, the authors do not provide information on tidal volume and hence minute ventilation, and no information is provided on metabolism (which are standard in the field of respiratory control). It is understood that not everything of interest (or potential interest) can reasonably be measured in any one study.

Our software did provide tidal volume (TV) and minute ventilation (MV) which was mentioned in the second last paragraph of the Fig 2 results section. No significant difference in mean values between sleep and wake or lesioned and control groups was observed, due to variability between mice. Source data are available.

My suggestion is to include a statement in the Discussion that a thorough assessment of respiratory control assessing other aspects of the respiratory phenotype would be interesting and likely informative. For example, it would be most interesting to determine if there is increased incidence of apnoea and indeed whether this relates to central and/or obstructive events. Interestingly, SpO₂ analysis suggests, as pointed out previously, that the lesioned mice express persistent hypoxaemia during recording sessions and not an intermittent hypoxaemia that might be expected with periods of intermittent apnoea (distinct from flow-limitation). It thus appears that the model principally elaborates a form of flow-limited sleep-disordered breathing, which in essence has been well characterised by the authors in the study.

We have included a statement in the Discussion as suggested, and have revised our terminology accordingly.

The revised Figure 2 is clear but the corresponding text Results section (Page 7) contains errors and some inexact language that can be easily remedied.

-Please refer to sex (not gender) throughout the manuscript.

-Page 7: I suggest referring to flow-limitation (not resistance), or clarify that you are referring to upper airway resistance and not bronchoconstriction. peak inspiratory "volume" (should be flow).

-"Surprisingly, breathing frequencywas shorter". Should be lower. The duration of one or more periods in the cycle was shorter.

-"Breath rate of 4s". Inexact. Rate is a value per unit time. So express as events per 4 s, or in other standard ways (i.e. per min).

-The example of 3 breaths per 2s is incorrect. Should be 5 for 150 breaths per min.

Thank you for these suggestions and corrections, which have been made.

A p value of $p < 0.0096$ is stated. Is the exact p value known? Should this be $p = 0.0096$?

The p value has been checked and corrected to $p = 0.021$.

The text states one-way t - test. Please clarify. One way ANOVA or t test? If t-test, state whether one or two-tailed throughout the manuscript. Confirm equal variance in data sets, or otherwise apply Welch test or equivalent.

This should have read a one-tailed unpaired t-test with Welch test. The precise statistical tests have now been stated throughout the manuscript. We confirm that where variance is not equal, a Welch test was applied, with this being stated in the Methods statistics section and in each figure legend.

Please carefully review this whole text section. It requires revision for exact expression. I appreciate that the nomenclature may be less familiar to the authors (indeed they are non-standard in the field of respiratory control), but better clarity of expression is required so that the findings are clearly stated matching the clear presentation of the findings in Fig. 2.

The text section reporting the findings of the breathing (Fig 2) has been reviewed and rewritten for clarity and precision of language. (pg 7 and 8)

It is evident that transitions between sleep states are increased in lesioned animals, and this is nicely illustrated in the revised Fig. 3.

The authors explain why it was not logistically easy to perform IH exposures and that collaborative

*studies were affected by COVID-related disruptions, which I fully accept.
Congratulations on the completion of a very interesting study.*

Thank you very much for your constructive suggestions to improve our study